

# Getting new algorithmic results by extending distance-hereditary graphs via split composition

Serafino Cicerone and Gabriele Di Stefano

Department of Information Engineering, Computer Science and Mathematics, University of L'Aquila, L'Aquila, Italy

## ABSTRACT

In this paper, we consider the graph class denoted as Gen(∗;P₃,C₃,C₅). It contains all graphs that can be generated by the split composition operation using path P₃, cycle C₃, and any cycle C₅ as components. This graph class extends the well-known class of distance-hereditary graphs, which corresponds, according to the adopted generative notation, to Gen(∗;P₃,C₃). We also use the concept of stretch number for providing a relationship between Gen(∗;P₃,C₃) and the hierarchy formed by the graph classes DH(k), with k ≥1, where DH(1) also coincides with the class of distance-hereditary graphs. For the addressed graph class, we prove there exist efficient algorithms for several basic combinatorial problems, like recognition, stretch number, stability number, clique number, domination number, chromatic number, and graph isomorphism. We also prove that graphs in the new class have bounded clique-width.

# INTRODUCTION

Distance-hereditary graphs have been introduced by (*Howorka 1977*), and are defined as those graphs in which every connected induced subgraph is isometric, that is the distance between any two vertices in the subgraph is equal to the one in the whole graph. Therefore, any connected induced subgraph of any distance-hereditary graph $G$ "inherits" its distance function from $G$. Formally:

**Definition 1** (*Howorka, 1977*) *A graph $G$ is a* distance-hereditary graph *if, for each connected induced subgraph $G'$ of $G$, the following holds: $d_{G'}(x,y) = d_G(x,y)$, for each $x,y \in G'$.*

This kind of graphs have been rediscovered many times (e.g., see *Bandelt & Mulder, 1986*). Since their introduction, dozens of papers have been devoted to them, and different kind of characterizations have been found: metric, forbidden subgraphs, cycle/chord conditions, level/neighborhood conditions, generative, and more (e.g., see *Brandstädt, Le & Spinrad, 1999*). Among such results, the generative properties resulted as the most fruitful for algorithmic applications, since they allowed researchers to efficiently solve many combinatorial problems in the context of distance-hereditary graphs

Corresponding author
Serafino Cicerone,
serafino.cicerone@univaq.it

(e.g., see *Brandstädt & Dragan, 1998*; *Chang, Hsieh & Chen, 1997*; *Gioan & Paul, 2007*; *Lin, Ku & Hsu, 2020*; *Nicolai & Szymczak, 2001*; *Rao, 2008a*).

From an applicative point of view, distance-hereditary graphs are mainly attractive due to their basic metric property. For instance, these graphs can model unreliable communication networks (*Cicerone, Di Stefano & Flammini, 2001*; *Esfahanian & Oellermann, 1993*) in which vertex failures may occur: at a given time, if sender and receiver are still connected, any message can be still delivered without increasing the length of the path used to reach the receiver.

Since in communication networks this property could be considered too restrictive, in *Cicerone & Di Stefano (2001)* the class of $k$–*distance-hereditary graphs* has been introduced. These graphs can model unreliable networks in which messages will eventually reach the destination traversing a path whose length is at most $k$ times the length of a shortest path computed in absence of vertex failures. The minimum $k$ a network guarantees regardless the failed vertices is called *stretch number*. Formally:

**Definition 2** (*Cicerone & Di Stefano, 1998*; *Cicerone & Di Stefano, 2001*) *Given a real number $k \geq 1$, a graph $G$ is a $k$–distance-hereditary graph if, for each connected induced subgraph $G'$ of $G$, the following holds: $d_{G'}(x,y) \leq k \cdot d_G(x,y)$, for each $x, y \in G'$.*

The class of all the $k$–distance-hereditary graphs is denoted by $\mathrm{DH}(k)$. Concerning this class of graphs, the following relationships hold:

- $\mathrm{DH}(1)$ coincides with the class of distance-hereditary graphs;
- $\mathrm{DH}(k_1) \subseteq \mathrm{DH}(k_2)$, for each $k_1 \leq k_2$.

Additional results about the class hierarchy $\mathrm{DH}(k)$, for any rational $k \geq 1$, can be found in *Cicerone (2011a)*, *Cicerone (2021)*, *Cicerone & Di Stefano (2000)*, *Cicerone & Di Stefano (2004)*. This hierarchy is *fully general*, that is, for each arbitrary graph $G$ there exists a number $k'$ such that $G \in \mathrm{DH}(k')$. It follows that the stretch number $s(G)$ of $G$ is the smallest rational number $t$ such that $G$ belongs to $\mathrm{DH}(t)$. In *Cicerone & Di Stefano (2001)*, it has been shown that stretch number $s(G)$ can be also computed as follows:

- the stretch number of any pair $\{u,v\}$ of distinct vertices is defined as $s_G(u,v) = D_G(u,v)/d_G(u,v)$, where $D_G(u,v)$ is the length of any longest induced path between $u$ and $v$, and $d_G(u,v)$ is the distance between the same pair of vertices;
- $s(G) = \max_{\{u,v\}} s_G(u,v)$.

It follows that if $G$ contains $n$ vertices, then $s(G) \leq \max\{1, (n-2)/2\}$. Interestingly, it has been shown that not all the possible rational numbers define possible stretch numbers. A positive rational number $t$ is called *admissible stretch number* if there exists a graph $G$ such that $s(G) = t$. The following result characterizes which numbers are admissible stretch numbers.

**Theorem 1** (*Cicerone & Di Stefano, 2000*; *Cicerone & Di Stefano, 2004*) *A rational number $t$ is an admissible stretch number if and only if $t \geq 2$ or $t = 2 - \frac{1}{i}$ for some integer $i \geq 1$.*

The class $\mathrm{DH}(2 - 1/i)$, for each $i > 1$, has been characterized in *Cicerone & Di Stefano (2004)* according to forbidden subgraphs. In *Cicerone (2011a)*, this characterization has

been generalized to the class containing any graph $G$ such that $s(G) < 2$. Since this class contains graphs with stretch number *strictly* less than two, it is denoted here as sDH(2). In *Cicerone (2011a)*, the class sDH(2) has been also characterized according to cycle-chord conditions.

In the literature, there are other hierarchies of classes that extend distance-hereditary graphs. The class DH($k, +$), for any integer $k \geq 0$, contains all graphs where, for each possible pair of vertices, the *difference* between the longest and the shortest induced paths between the vertices is bounded by an integer value $k$ (in other words, in each possible connected induced subgraph, the distance may increase up to an *additive* factor $k$). This hierarchy has been independently introduced in *Cicerone, D'Ermiliis & Di Stefano (2001)* and *Aïder (2002)*, and further results can be found in *Cicerone & Di Stefano (2003)* and *Rautenbach (2004)*. Another hierarchy has been proposed in *Cicerone & Di Stefano (1999a)* and *Cicerone & Di Stefano (1999b)*. It extends the distance-hereditary graphs by exploiting the well-known characterization based on "one-vertex extension" operations. Accordingly, each distance-hereditary graph can be generated starting from a single vertex and by applying $\beta$ and $\gamma$, that add *false* and *true twins* respectively, and $\alpha$, that adds a pendant node (i.e., the smallest bipartite graphs). In this hierarchy, $\Phi^0$ denotes the class of distance-hereditary graphs, and $\Phi^i$, for each integer $i \geq 1$, denotes the class containing all the graphs that can be generated by applying $\beta$, $\gamma$, and $\alpha$, where the latter now can append any bipartite graphs in $\Phi^{i-1}$. It follows that the class of parity graphs (*Burlet & Uhry, 1984*) represents a limit for $\Phi^i$.

## Motivation and results

One of the main motivations for the introduction of the DH($k$) hierarchy was to refine the borderline between the classes of complexity P and NP. In particular, the initial class DH(1) was found to have multiple characterizations that, when properly exploited, produced polynomial solutions for several combinatorial problems which are NP complete in the class of general graphs. So, the idea with the introduced hierarchy was to extend, if possible, some of these polynomial results to a class DH($k$) for some constant $k > 1$. Despite some interesting properties found for the generic DH($k$) class, this ambitious goal was not achieved. As a consequence, a slightly different approach was pursued in *Cicerone (2011b)*: to define a new class which at the same time was a superclass of DH(1) and with evident relationships with the DH($k$) hierarchy.

The introduced class has been denoted as $Gen(*; P_3, C_3, C_5)$ and it contains all graphs that can be *generated* by applying the split composition operation (*Cunningham, 1982*) using path $P_3$, cycle $C_3$, and any cycle $C_5$ as components. We use the notation $Gen(\texttt{opn}; \texttt{list\_of\_components})$[1] for classes whose elements can be defined in a generative way. In *Cicerone (2011b)* it is stated (but with omitted proofs) that in the class $Gen(*; P_3, C_3, C_5)$ several problems that are in general NP-hard can be efficiently solved. In particular:

1. $Gen(*; P_3, C_3, C_5)$ is a proper subclass of sDH(2);
2. there exist linear time algorithms for the recognition problem, the graph isomorphism problem, and for computing the stretch number;

[1] The syntax *Gen*(opn; list_of_components) emphasizes the generative definition of graphs in the class, like *Forb*(list_of_subgraphs) is often used to define graph classes according to a list of minimal forbidden subgraphs.

3. by exploiting a result provided by *Rao (2008b)*, it can be derived that there exist linear time algorithms for computing the stability number, clique number, domination number and its variants, and an $O(n^3)$-time algorithm for computing the chromatic number; it is also possible to prove that all graphs in $Gen(*; P_3, C_3, C_5)$ have bounded clique-width.

All the above algorithmic results are obtained by exploiting in some way the generative definition of each graph belonging to $Gen(*; P_3, C_3, C_5)$ according to the classical decomposition approach: the problem at hand is first solved in each component and then the solutions obtained in each component are composed/manipulated to get the solution for the whole graph.

In this work we provide the following contribution. First, we show a counterexample for a theorem given in *Cicerone (2011b)*; the property contained in that theorem was the basis of the algorithm for calculating the stretch number of graphs belonging to the studied class. Then, we provide new structural properties for graphs in sDH(2), and we use such properties for devising a new technique for computing the stretch number. This technique in itself is simple (dynamic programming on the split decomposition tree), but it contains a very interesting way to store the output of subproblems that might find applications elsewhere, i.e., for other resolution techniques based on split decomposition. Finally, we provide full proofs of all the other results claimed in *Cicerone (2011b)*.

## Outline

The paper is organized as follows. In 'Notation, Basic Concepts, and Preliminary Results' we give basic notation, necessary concepts concerning the split decomposition, and additional concepts and results concerning the stretch number. In 'Extending Distance-Hereditary Graphs Via Split Composition' we formally recall the class $Gen(*; P_3, C_3, C_5)$ and show that the recognition problem for this new class can be solved in linear time. 'Structural Properties About Graphs in sDH(2)' provides some properties about graphs in sDH(2) that are used in 'Computing the Stretch Number' to provide a linear time algorithm for computing the stretch number in any graph belonging to the newly defined class. In 'Other Combinatorial Problems' we observe how known results can be exploited to solve many other combinatorial problems in the new class. Finally, 'Conclusion' provides some concluding remarks.

## NOTATION, BASIC CONCEPTS, AND PRELIMINARY RESULTS

We consider finite, simple, loop-less, undirected and unweighted graphs $G = (V, E)$ with vertex set $V$ and edge set $E$. A *subgraph* of $G$ is a graph having all its vertices and edges in $G$. Given $S \subseteq V$, the *induced subgraph* $G[S]$ of $G$ is the maximal subgraph of $G$ with vertex set $S$. Given $u \in V$, $N_G(u)$ denotes the set of *neighbors* of $u$ in $G$, and $N_G[u] = N_G(u) \cup \{u\}$.

A sequence of pairwise distinct vertices $(x_0, x_1, \ldots, x_k)$ is a *path* in $G$ if $(x_i, x_{i+1}) \in E$ for $0 \leq i < k$. A *chord* of a path is any edge joining two non-consecutive vertices in the path, and a path is an *induced path* if it has no chords. Each vertex $x_i$, $0 < i < k$, is an *internal vertex* of the path $(x_0, x_1, \ldots, x_k)$. We denote by $P_k$ any path with $k \geq 3$ vertices. Two vertices

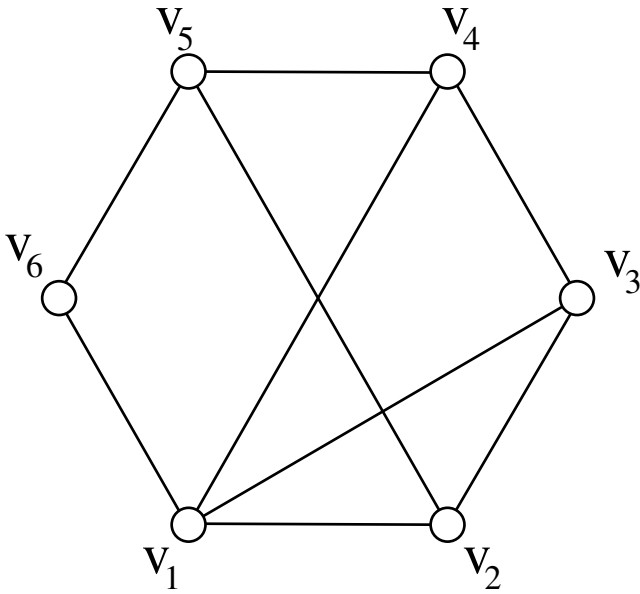

**Figure 1** The chord distance of of this $C_6$ graph is 2 because vertices $v_1$ and $v_2$ are consecutive and every chord is incident to them, and there is no other set with less than two vertices with the same property.

$x$ and $y$ are *connected* in $G$ if there exists a path $(x, \dots, y)$ in $G$. A graph is *connected* if every pair of vertices is connected.

A *cycle* in $G$ is a path $(x_0, x_1, \dots, x_{k-1})$ where also $(x_0, x_{k-1}) \in E$. Two vertices $x_i$ and $x_j$ are *consecutive* in $C_k$ if $j = (i+1) \bmod k$ or $i = (j+1) \bmod k$. A *chord* of a cycle is an edge joining two non-consecutive vertices in the cycle. We denote by $C_k$ any cycle with $k \geq 3$ vertices, whereas $H_k$ denotes a *hole*, i.e., a cycle $C_k$, $k \geq 5$, without chords. The *chord distance* of a cycle $C_k$ is denoted by $cd(C_k)$, and it is defined as the minimum number of consecutive vertices in $C_k$ such that every chord of $C_k$ is incident to some of such vertices (see Fig. 1 for an example of chord distance). We assume $cd(H_k) = 0$.

A *clique* (or *complete graph*) is any graph in which every two distinct vertices are adjacent; a clique with $n$ vertices is denoted by $K_n$ (but notice we use $C_3$ to denote the clique with three vertices). A *complete bipartite graph* is a graph whose vertices can be partitioned into two subsets $V_1$ and $V_2$ such that no edge has both endpoints in the same subset, and every possible edge that could connect vertices in different subsets is part of the graph. Symbol $K_{n,m}$ is used to denote a complete bipartite graph with $|V_1| = n$ and $|V_2| = m$. A *star* is any graph $K_{1,n}$ with $n \geq 2$.

The length of any shortest path between two vertices $x$ and $y$ in a graph $G$ is called *distance* and is denoted by $d_G(x, y)$. Moreover, the length of any longest induced path between them is denoted by $D_G(x, y)$. If $x$ and $y$ are distinct vertices, we use the symbols $p_G(x, y)$ and $P_G(x, y)$ to denote a shortest and a longest induced path between $x$ and $y$, respectively. Sometimes, when no ambiguity occurs, we also use $p_G(x, y)$ and $P_G(x, y)$ to denote the sets of vertices belonging to the corresponding paths. If $d_G(x, y) \geq 2$, then $\{x, y\}$ is a *cycle-pair*

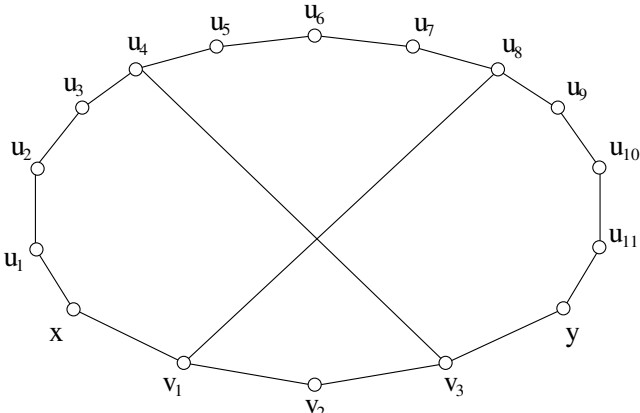

**Figure 2** **An example showing that Lemma 2 does not hold in graphs having stretch number greater than 2.** Vertex $v_2$ is internal to a shortest path between two vertices that provides the stretch number but it is not incident to any chord.

if there exist two induced paths $p_G(x,y)$ and $P_G(x,y)$ such that $p_G(x,y) \cap P_G(x,y) = \{x,y\}$. In other words, if $\{x,y\}$ is a cycle-pair, then the vertices in $p_G(x,y) \cup P_G(x,y)$ induce a cycle in $G$; this cycle is denoted by $G[x,y]$. In Fig. 1, $\{v_3,v_6\}$ is a cycle-pair that induces the cycle $(v_3,v_4,v_5,v_6,v_1)$; in particular, $G[v_3,v_6]$ is induced by $p_G(v_3,v_6) = (v_3,v_1,v_6)$ and $P_G(v_3,v_6) = (v_3,v_4,v_5,v_6)$. The following lemma states that cycle-pairs are useful to determine the stretch number.

**Lemma 1** (*Cicerone & Di Stefano, 2001*) *If $G$ is a graph such that $s(G) > 1$, then there exists a cycle-pair $\{u,v\}$ that induces the stretch number of $G$, that is $s(G) = s_G(u,v)$.*

As a consequence of this lemma, if $\{u,v\}$ is a cycle-pair that induces the stretch of $G$ then the cycle $G[u,v]$ is called *inducing-stretch cycle* for $G$. In Fig. 1, the represented graph $G$ belongs to DH(3/2); moreover both $G[v_3,v_6]$ and $G[v_3,v_5]$ are inducing-stretch cycles for $G$.

The following lemma refers to any graph $G$ belonging to sDH(2) \ DH(1) and provides a very useful property concerning the chords of any induced stretch-cycle of $G$.

**Lemma 2** (*Cicerone & Di Stefano, 2004*) *Let $G$ be a graph such that $1 < s(G) < 2$, and let $G,[y]$ any inducing-stretch cycle of $G$. Then, either $G[x,y]$ is chordless or each internal vertex of $p_G(x,y)$ is incident to a chord of $G[x,y]$.*

Unfortunately this property does not hold for graphs $G$ with $s(G) > 2$. For instance, Fig. 2 shows a graph with stretch number equal to 3 where: (1) the cycle $G[x,y]$ generated by $P(x,u_1,u_2,\ldots,u_{11},y)$ and $p(x,v_1,v_2,v_3,y)$ is an induced-stretch cycle, and (2) there exists $v_2$ which is an internal vertex of $p_G(x,y)$ but it is not incident to any chord of $G[x,y]$.

We conclude this section by introducing a notion related to chords. Let $C = (x,u_1,u_2,\ldots,u_m,y,v_n,v_{n-1},\ldots,v_1)$ be a cycle with $m \geq n$, and assume that each chord of $C$ (if any) is incident to some vertex $v_j$, for $1 \leq j \leq m$. Given $v_j$, $1 \leq j \leq n$, then $u_{l_j}$ and $u_{r_j}$ denote the vertices incident to the *leftmost* and *rightmost* chord of $v_j$, respectively. Formally,

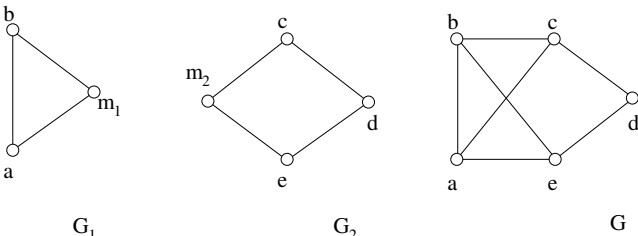

**Figure 3** An example of split composition: $G = (G_1, m_1) * (G_2, m_2)$.

- $l_j = \min\{j' | 1 \leq j' \leq m \text{ and } (v_j, u_{j'}) \text{ is a chord of } C\}$
- $r_j = \max\{j' | 1 \leq j' \leq m \text{ and } (v_j, u_{j'}) \text{ is a chord of } C\}$

If $v_j$ is not incident to any chord of $C$, we say that $r_j$ and $l_j$ are undefined.

## Split composition and decomposition

In this section we briefly recall the concepts of *split composition* and *decomposition* as defined by *Cunningham (1982)*. Let $G_1$, $G_2$ be two graphs having disjoint vertex sets $V_1 \cup \{m_1\}$, $V_2 \cup \{m_2\}$ and edge sets $E_1, E_2$, respectively. The split composition of $G_1$ and $G_2$ with respect to the *joining vertices* $m_1$ and $m_2$ is the graph $G$ having vertex set $V = V_1 \cup V_2$ and edge set $E = E_1' \cup E_2' \cup \{(u,v) | u \in N_{G_1}(m_1), v \in N_{G_2}(m_2)\}$, where $E_i' = \{(u,v) \in E_i | u, v \in V_i\}$ for $i = 1, 2$.

The composition is denoted as $G = (G_1, m_1) * (G_2, m_2)$, or simply $G = G_1 * G_2$ when we are not interested in the joining vertices. An example of split composition is shown in Fig. 3.

The split composition has an inverse operation, namely the *split decomposition*. Let $G_1$, $G_2$ be two graphs having disjoint vertex sets $V_1 \cup \{m_1\}$, $V_2 \cup \{m_2\}$ and edge sets $E_1, E_2$, respectively. If $G = (G_1, m_1) * (G_2, m_2)$ and $|V_1|, |V_2| \geq 2$, then we say that $\{G_1, G_2\}$ is a *simple decomposition* of G. We call $\{V_1, V_2\}$ the *split* of $G$ associated with the simple decomposition $\{G_1, G_2\}$, being $m_1$ and $m_2$ the associated joining vertices (these vertices are often referred as "marked" vertices when they are introduced by the split decomposition operation - cf Fig. 4). If $G$ has a split we say that $G$ is *split-decomposable*. The *split decomposition* of a graph $G$ is the set $\mathcal{D}(G)$ of graphs obtained by the following recursive procedure:

- if $G$ has a split $\{V_1, V_2\}$, then apply the split decomposition to graphs $G_1$ and $G_2$ obtained by the simple decomposition $\{G_1, G_2\}$;
- if $G$ does not have a split then $G$ is called *prime*.

Each element of $\mathcal{D}(G)$ is called *component*. Sometimes it is useful to associate a *decomposition tree* $T(G)$ to $\mathcal{D}(G)$ as follows: each vertex of $T(G)$ corresponds to a prime components in $\mathcal{D}(G)$; furthermore two vertices of $T(G)$ are adjacent if and only if the corresponding components have been obtained by a simple decomposition. An important property of the split decomposition is that every component of $\mathcal{D}(G)$ is isomorphic to an induced subgraph of $G$. For example, the graph $G_2$ of Fig. 3 is isomorphic to the subgraph induced by the vertex set $\{a, c, d, e\}$.

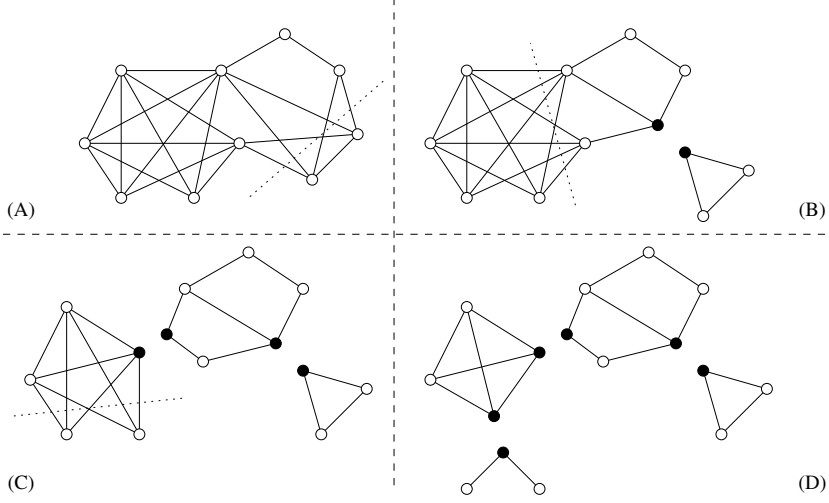

**Figure 4** **From (A) to (D), in order, the sequence of splits to get the Cunningham decomposition $\mathcal{D}(G)$ of a graph G.** Dotted lines show the splits performed at each step, black vertices represent joining/marked vertices. It is easy to see that the corresponding decomposition tree $T(G)$ is a $P_4$ graph.

The split decomposition of a graph is not necessarily unique, but Cunningham proved the following uniqueness result:

**Theorem 2** (*Cunningham, 1982*) *Each connected graph has a unique split decomposition into prime graphs, stars, and cliques with a minimum number of components.*

**Remark 1** *The decomposition used in the previous theorem, called* Cunningham decomposition*, is not necessarily a split decomposition, since stars and cliques are split-decomposable. It is easy to find a minimal split decomposition from an unique split decomposition, or vice versa (Cunningham, 1982). See* Fig. 4 *for an example of Cunningham decomposition of a graph.*

All known algorithms that compute the split decomposition compute in fact a Cunningham decomposition. The first algorithm was given by Cunningham and has running time $O(n^3)$ (*Cunningham, 1982*). This result has been improved to $O(nm)$ in *Gabor, Supowit & Hsu (1989)*, and to $O(n^2)$ in *Ma & Spinrad (1994)*. Finally, Dahlhaus has given a linear time algorithm in *Dahlhaus (2000)*. Recently, more practical algorithms have been proposed (*Charbit, De Montgolfier & Raffinot, 2012*; *Gioan et al., 2014*).

From now on, unless otherwise specified, when we use the term "split decomposition" we always refer to the Cunningham decomposition.

Some graph classes are nicely decomposable by split decomposition. A graph is a *distance-hereditary graph* if and only if every graph in its split decomposition is a star or a clique (*Cicerone & Di Stefano, 1999a*). A graph is a *parity graph* if and only if every prime graph in its split decomposition is bipartite or a clique (*Cicerone & Di Stefano, 1999b*). A graph is a *circle graph* if and only if every prime graph in its split decomposition is a circle graph (*Gabor, Supowit & Hsu, 1989*). The best known recognition algorithms for circle graphs (*Spinrad, 1994*) and parity graphs (*Cicerone & Di Stefano, 1999b*; *Dahlhaus, 2000*)

are based on split decomposition. Moreover, a graph is perfect if and only if every prime graph in the split decomposition is perfect (*Bixby, 1984*).

**Split composition and stretch number**

In the remainder we will often use the following basic property - which directly follows from the definition of split composition, and the subsequent lemma.

**Lemma 3** *Let* $G = (G_1, m_1) * (G_2, m_2)$. *If* $u \in G_1 \setminus N_{G_1}[m_1]$ *and* $v \in G_2 \setminus N_{G_2}[m_2]$ *then the following properties hold:*

- $d_G(u, v) = d_{G_1}(u, m_1) + d_{G_2}(v, m_2) - 1$;
- $D_G(u, v) = D_{G_1}(u, m_1) + D_{G_2}(v, m_2) - 1$.

**Proof** The former trivially follows from to the definition of split composition. Concerning the latter, it is less trivial but it depends on the fact that $G$ contains all the possible edges between vertices in $N_{G_1}(m_1)$ and vertices in $N_{G_2}(m_2)$. Hence, $P_G(u, v)$ cannot include more than two vertices in $N_{G_1}(m_1) \cup N_{G_2}(m_2)$, otherwise it would not be an induced path. □

**Lemma 4** *Let* $G_1 = (V_1, E_1)$ *and* $G_2 = (V_2, E_2)$ *be two graphs. If* $G = (G_1, m_1) * (G_2, m_2)$, *then*

$$s(G) = \max\left\{ s(G_1), s(G_2), \max_{u \in V_1 \setminus N_{G_1}[m_1], v \in V_2 \setminus N_{G_2}[m_2]} \left\{ \frac{D_{G_1}(u, m_1) + D_{G_2}(v, m_2) - 1}{d_{G_1}(u, m_1) + d_{G_2}(v, m_2) - 1} \right\} \right\}.$$

**Proof** If we assume that vertices $u$ and $v$ gives the stretch number of $G$, that is $s(G) = s_G(u, v)$, then three different cases may occur:

- both $u$ and $v$ are in $V_1$;
- both $u$ and $v$ are in $V_2$;
- $u$ belongs to $V_1 \setminus N_{G_1}[m_1]$ and $v$ belongs to $V_2 \setminus N_{G_2}[m_2]$.

In the first case, we get $s(G) = s(G_1)$, as any path between $u$ and $v$ containing more than one vertex in $V_2$ is not induced. Similarly for the second case where $s(G) = s(G_2)$. In the last case, the statement follows from Lemma 3. □

## EXTENDING DISTANCE-HEREDITARY GRAPHS VIA SPLIT COMPOSITION

In this section we define a new graph class which represents a superclass of distance-hereditary graphs.

One of the most popular characterizations of distance-hereditary graphs is based on *one-vertex extension* operations (*Bandelt & Mulder, 1986*). More precisely, if $G$ is a graph and $u$ any vertex of $G$, then the one-vertex operations that can be applied to $u$ to extend $G$ are the following:

- $\alpha(G, u; v)$ adds a new vertex $v$ to $G$ and makes it adjacent only to $u$;
- $\beta(G, u; v)$ adds a new vertex $v$ to $G$ and makes it adjacent to every neighbor of $u$;

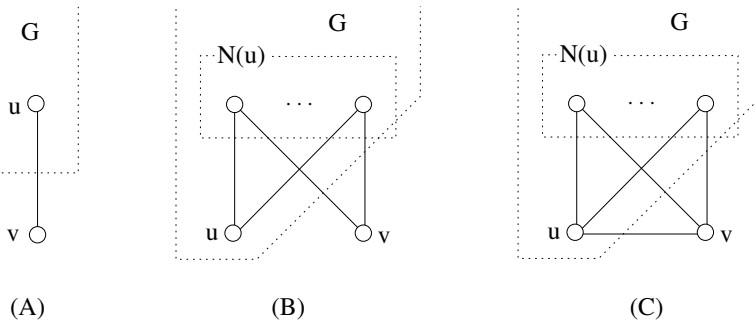

**Figure 5** The one-vertex extension operations that generate each distance-hereditary graph: (A), (B), and (C) show $\alpha(G, u; v)$, $\beta(G, u; v)$, and $\gamma(G, u; v)$, respectively.

-  $\gamma(G, u; v)$ adds a new vertex $v$ to $G$ and makes it adjacent to $u$ and to every neighbor of $u$.

See Fig. 5 for a graphical representation of these operations. As stated in the following theorem, any distance-hereditary graph can be obtained by exploiting the operations $\alpha$, $\beta$ and $\gamma$.

**Theorem 3** (*Bandelt & Mulder, 1986*) *Every distance-hereditary graph is obtained starting from a single vertex and by applying a proper sequence of operations  $\alpha$, $\beta$, and  $\gamma$.*

We now observe that this result can be reformulated in terms of a single operation, namely, the split composition.

**Lemma 5** *A graph G is distance-hereditary if each of its connected components has at most two vertices or it is obtained by applying the split composition and using path  $P_3$ and cycle $C_3$ as components.*

**Proof** It is easy to verify that the following relationships hold:
-  $\alpha(G, u; v) \equiv (G, u) * (P_3, v)$, where $v$ is an external vertex in the path $P_3$;
-  $\beta(G, u; v) \equiv (G, u) * (P_3, v)$, where $v$ is an internal vertex in the path $P_3$;
-  $\gamma(G, u; v) \equiv (G, u) * (C_3, v)$, where $v$ is any vertex in the cycle $C_3$.

The proof is concluded by observing that these relationships also show that *every* vertex of the components $P_3$ and $C_3$ can be used as joining vertex.   □

Notice that the previous result cannot be directly derived from the well known result that decomposing a distance-hereditary graph by means of the split decomposition results in components with at most 3 vertices (*Cicerone & Di Stefano, 1999a*; *Hammer & Maffray, 1990*). In fact, such result does not provide information about how such components are joined.

From the above Lemma it follows that the class of distance-hereditary graphs can be denoted as $Gen(*; P_3, C_3)$. Now, extending $Gen(*; P_3, C_3)$ via split composition is just

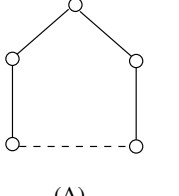 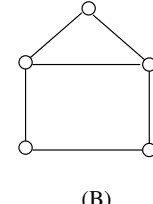 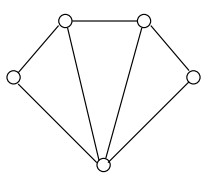 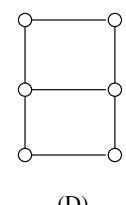

(A)  (B)  (C)  (D)

**Figure 6  The minimal forbidden subgraphs of distance-hereditary graphs: from (A) to (D), in order, the hole $H_n$ with $n \geq 5$, the house, the fan, and the domino.** Dashed lines represent paths of length at least one.

matter of selecting a new component to be used along with $P_3$ and $C_3$. To this aim we recall the following additional characterization of distance-hereditary graphs (see Fig. 6).

**Theorem 4** (*Bandelt & Mulder, 1986*) *A graph G is a distance-hereditary graph if and only if it does not contain, as induced subgraph, the following graphs: the hole $H_n$, $n \geq 5$, the house, the fan, and the domino.*

This result states that only few kinds of graphs cannot appear as induced subgraphs in a distance-hereditary graph: the hole $H_n$, $n \geq 6$, cycles $C_5$ with chord distance at most one (which correspond exactly to $H_5$, the house, and the fan), and the domino. Since the smallest ones among such forbidden subgraphs are the cycles $C_5$ with chord distance at most one, then the following definition formalizes the new graph class:

**Definition 3** *A graph G belongs to $Gen(*; P_3, C_3, C_5)$ if each of its connected components has at most two vertices or it is obtained by means of the split composition using path $P_3$, cycle $C_3$, and any cycle $C_5$ as components.*

For sake of simplicity, in this definition we do not limit the chord distance of cycles $C_5$ used as components, and hence any cycle with five vertices can be used. Anyway, notice that if a cycle $C_5$ with chord distance more than one is used, this corresponds to join a distance-hereditary graph, which in turn is formed by $P_3$ and $C_3$ components. In conclusion, a single join operation with a cycle $C_5$ with chord distance more than one corresponds to join a sequence of $P_3$ and $C_3$ components.

By comparing Lemma 5 and Definition 3 it follows that

$$DH(1) \subsetneq Gen(*; P_3, C_3, C_5).$$

The class DH(1) is hereditary. In general, a class $\mathcal{C}$ is hereditary if any induced subgraph of a graph in $\mathcal{C}$ is in $\mathcal{C}$. We show that this nice property is also valid for the new class $Gen(*; P_3, C_3, C_5)$.

**Theorem 5** $Gen(*; P_3, C_3, C_5)$ *is a hereditary class.*

**Proof** Let $G \in Gen(*; P_3, C_3, C_5)$, $v$ a vertex of $G$, and $G'$ the subgraph obtained by removing $v$ from $G$. If $G$ has less than three nodes, the proof is trivial. In the remaining cases, according to Definition 3, there exists a generative sequence of components $B_1, B_2, \ldots, B_t$, $t \geq 1$, for $G$. This means:

- $B_i \in \{P_3, C_3, C_5\}$, $1 \leq i \leq t$;

- $G_1 \equiv B_1$;
- $G_i \equiv G_{i-1} * B_i$, $2 \leq i \leq t$;
- $G \equiv G_t$.

We prove the statement by providing a generative sequence for $G'$. Assume that $B_i$ is the component such that $v \in B_i$. According to $B_i$, there are different generative sequences for $G'$ (cf. Fig. 7):

- $B_i \equiv C_3$. The generative sequence for $G'$ is $B_1, B_2, \ldots, B_{i-1}, B_{i+1}, \ldots, B_t$ (cf. graph $G_1 \setminus \{v_9\}$ in Fig. 7);
- $B_i \equiv P_3$ and $v$ is an external vertex of the path $P_3$. As in the previous case, the generative sequence for $G'$ is $B_1, B_2, \ldots, B_{i-1}, B_{i+1}, \ldots, B_t$ (cf. graph $G_1 \setminus \{v_4\}$ in Fig. 7);
- $B_i \equiv P_3$ and $v$ is an internal vertex of the path $P_3$. Then $v$ is an articulation point of $G$, and hence $G'$ is disconnected into two or more connected subgraphs. It can be observed that removing from $G$ all the edges adjacent to $v$ may affect many components $B_i$, but each affected $B_i$ reduces its size and either it is no longer needed as a components for building $G'$ or it is splitted in smaller components, as in the first analyzed case (cf. graph $G_2 \setminus \{v_9\}$ in Fig. 7);
- $B_i \equiv C_5$. Let $B_i'$ be the subgraph obtained from $B_i$ by removing $v$. Since $B_i'$ has 4 vertices, then $B_i'$ is a distance-hereditary graph, and hence the minimal split decomposition of $B_i'$ contains only $P_3$ and $C_3$ components. Hence, the generative sequence for $G'$ is given by $B_1, B_2, \ldots, B_{i-1}$ followed by the components in $\mathcal{D}(B_i')$ and then followed by $B_{i+1}, \ldots, B_t$.

This concludes the proof. □

## Recognition problem

The recognition problem for the new class can be formulated as follows: given a graph $G$, decide whether $G$ belongs to $Gen(*; P_3, C_3, C_5)$. After proving a useful lemma, we will show that it can be easily solved by using split decomposition algorithms.

**Lemma 6** *If $G \in Gen(*; P_3, C_3, C_5)$ then every prime graph in $\mathcal{D}(G)$ is a cycle $C_5$ with chord distance at most one.*

**Proof** Let $B$ be a prime component of $\mathcal{D}(G)$. Since $B$ is an induced subgraph of $G$, by Theorem 5 it follows that $B \in Gen(*; P_3, C_3, C_5)$ and hence there exists a generative sequence $B_1, B_2, \ldots, B_t$, $t \geq 1$, for $B$. If $|B| \geq 6$ then the generative sequence contains at least two components (i.e., $t \geq 2$): this is a contradiction for $B$ prime. Hence $t = 1$, i.e., the generative sequence is composed by $B$ exactly, and hence $B \in \{P_3, C_3, C_5\}$. Since $B$ is prime, it cannot be $P_3$ (which is a star) and $C_3$ (which is a clique). Then it follows that $B$ is a cycle $C_5$. This cycle cannot have chord distance more than one otherwise it would be a distance-hereditary graph and hence split-decomposable into $P_3$ and $C_3$ components. □

**Theorem 6** *A graph $G$ belongs to $Gen(*; P_3, C_3, C_5)$ if and only if each component in $\mathcal{D}(G)$ is a star, a clique, or a cycle $C_5$ with chord distance at most one.*

**Proof** ($\Rightarrow$) It directly follows by using Lemma 6. ($\Leftarrow$) Let us assume that each component of $\mathcal{D}(G) = \{B_1, B_2, \ldots, B_n\}$ is a star, a clique, or a cycle $C_5$ with chord distance at most one. Now, perform the following operations on each component $B_i$, $1 \leq i \leq n$, which is a clique or a star:

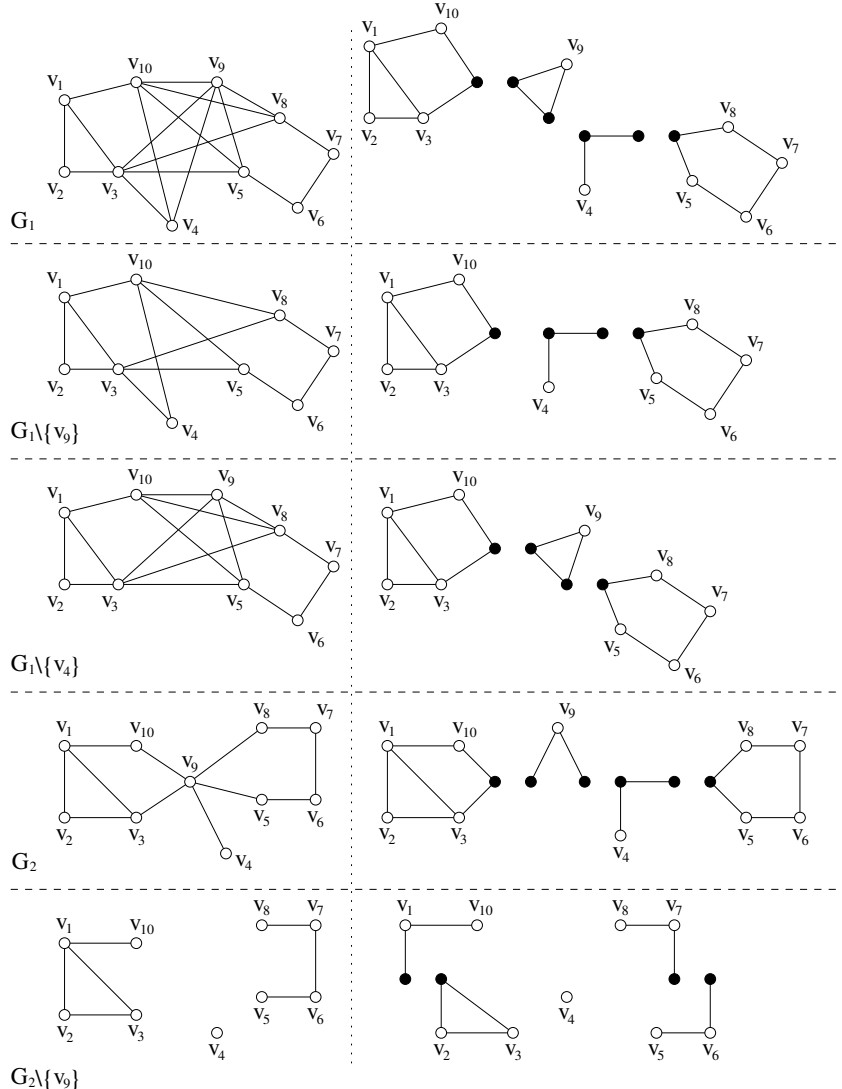

**Figure 7  Visualization of some arguments for the proof of Theorem 5.**

- if $B_i$ is a clique $K_t$, $t \geq 3$, then replace $K_t$ by $C_3 * C_3 * \cdots * C_3$ with $t-2$ cliques $C_3$ according to Fig. 8;
- if $B_i$ is a star $K_{1,t}$, $t \geq 2$, then replace $K_{1,t}$ by $P_3 * P_3 * \cdots * P_3$ with $t-1$ paths $P_3$ according to Fig. 9.

From components $B_1, B_2, \ldots, B_n$ and their modifications we get a split decomposition with components in $\{P_3, C_3, C_5\}$. From this split decomposition it is easy to get a generative sequence showing that $G \in Gen(*; P_3, C_3, C_5)$. $\square$

**Theorem 7** *The recognition problem for the class $Gen(*; P_3, C_3, C_5)$ can be solved in linear time.*

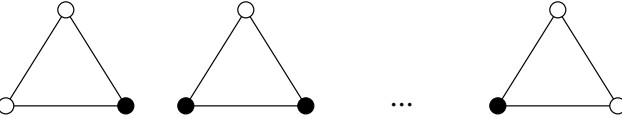

**Figure 8** Decomposing a clique $K_t$, $t \geq 3$, into $C_3 * C_3 * \cdots * C_3$ with $t-2$ cliques $C_3$ (nearby black vertices represent pairs of joining vertices).

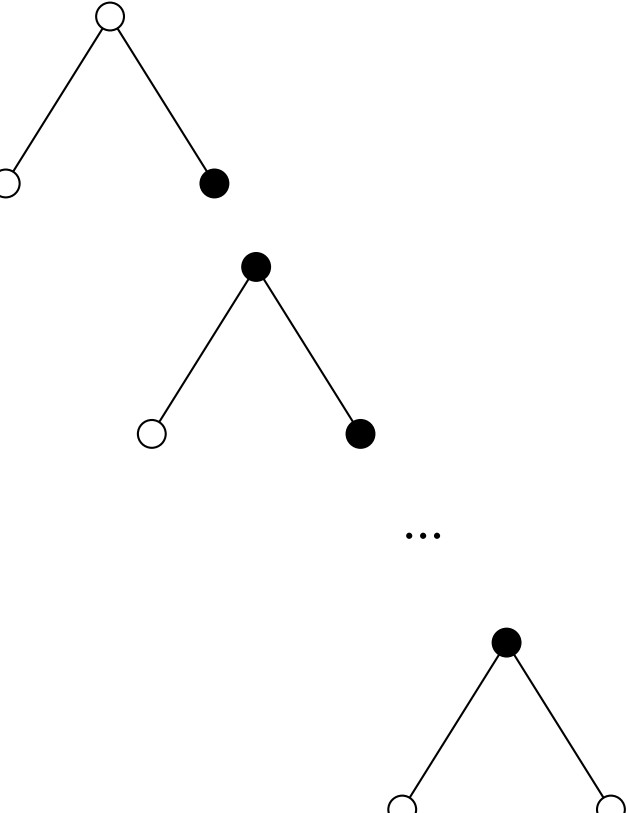

**Figure 9** Decomposing a star $K_{1,t}$, $t \geq 2$, into $P_3 * P_3 * \cdots * P_3$ with $t-1$ paths $P_3$ (nearby black vertices represent pairs of joining vertices).

**Proof** Let $G$ be a graph in $Gen(*; P_3, C_3, C_5)$. The statement directly follows from Theorem 6 and from the linear time algorithm proposed in *Dahlhaus (2000)* for computing the Cunningham decomposition $\mathcal{D}(G)$. □

## STRUCTURAL PROPERTIES ABOUT GRAPHS IN SDH(2)

In this section we provide some useful properties about graphs in sDH(2) (we remind that this class contains any graph $G$ such that $s(G) < 2$). In particular, we first show that $Gen(*; P_3, C_3, C_5)$ extends DH(1) till to sDH(2). Then, by recalling that DH(1) is characterized by split decomposition (i.e., each distance-hereditary graph is split decomposable into cliques and stars), we ask whether the same happens to graphs in

sDH(2). In this respect, we show that "the core" of each graph $G \in$ sDH(2) can be fully decomposed into a sequence of cycles $C_5$, where by core we informally mean an induced subgraph $G[u,v]$ where $\{u,v\}$ is a pair of vertices such that $s(G) = s_G(u,v)$.

We start by proving that sDH(2) is closed under split composition.

**Lemma 7** *Let $G_1$ and $G_2$ two graphs both belonging to sDH(2). If $G = G_1 * G_2$ then $G \in$ sDH(2).*

**Proof** Assume that $G_1 = (V_1, E_1)$, $G_2 = (V_2, E_2)$, and $G = (G_1, m_1) * (G_2, m_2)$. According to Lemma 4, to show that the statement holds, it is sufficient to prove that

$$\frac{D_{G_1}(u, m_1) + D_{G_2}(v, m_2) - 1}{d_{G_1}(u, m_1) + d_{G_2}(v, m_2) - 1} < 2$$

for each $u \in V_1 \setminus n[m_1]$ and $v \in V_2 \setminus N_{G_2}[m_2]$. Since $D_{G_1}(u, m_1)/d_{G_1}(u, m_1) < 2$ and $D_{G_2}(m_2, v)/d_{G_2}(m_2, v) < 2$ hold by hypothesis, then $D_{G_1}(u, m_1) \leq 2d_{G_1}(u, m_1) - 1$ and $D_{G_2}(m_2, v) \leq 2d_{G_2}(m_2, v) - 1$. Hence $D_{G_1}(u, m_1) + D_{G_2}(m_2, v) - 1 \leq (2d_{G_1}(u, m_1) - 1) + (2d_{G_2}(m_2, v) - 1) - 1 = 2d_{G_1}(u, m_1) + 2d_{G_2}(m_2, v) - 3 < 2d_{G_1}(u, m_1) + 2d_{G_2}(m_2, v) - 2.$ □

**Corollary 1** *Let $G$ be a graph. Then $G \in$ sDH(2) if and only if $B \in$ sDH(2) for each prime components $B$ of $\mathcal{D}(G)$.*

**Proof** Let us assume $G \in$ sDH(2). Since each prime components $B$ of $\mathcal{D}(G)$ is an induced subgraph of $G$, then $s(B) < 2$. Assume now that $B \in$ sDH(2) for each prime components $B$ of $\mathcal{D}(G)$. Concerning the components that are not prime, observe that both stars and cliques have stretch number equal to one. Then, by Lemma 7 the resulting graph $G$ obtained by the composition of all the components of $\mathcal{D}(G)$ is such that $G \in$ sDH(2). □

Lemma 7 allow us to give a clear relationship between the new graph class $Gen(*; P_3, C_3, C_5)$ and the class hierarchy DH($k$), $k \geq 1$.

**Corollary 2** DH(1) $\subsetneq Gen(*; P_3, C_3, C_5) \subsetneq$ DH(2).

**Proof** DH(1) $\subsetneq Gen(*; P_3, C_3, C_5)$ follows from Definitions 2 and 3. If $G \in Gen(*; P_3, C_3, C_5)$, Theorem 6 and Lemma 7 imply $s(G) < 2$, and hence $Gen(*; P_3, C_3, C_5) \subsetneq$ DH(2). □

Concerning properties about the split decomposition of graphs in sDH(2), it is worth to observe that there exist prime graphs in sDH(2) which are larger than the components used to generate graphs in $Gen(*; P_3, C_3, C_5)$. Figure 10A shows a prime graph with stretch number 3/2 and not belonging to $Gen(*; P_3, C_3, C_5)$; in Fig. 10B we show that it is possible to find graphs with the same properties but with any size.

Let $i \geq 2$ be an integer, and let $G$ be a graph such that $s(G) = 2 - \frac{1}{i}$. Then, by definition of stretch number, there exist vertices $x, y \in G$ such that

$$s(G) = s_G(x,y) = \frac{D_G(x,y)}{d_G(x,y)} = \frac{2i-1}{i}.$$

Since integers $2i - 1$ and $i$ are coprime, then there exists an inducing-stretch cycle $G[x,y]$ having at least $(2i - 1 + 1) + (i + 1) - 2 = 3i - 1$ vertices. We say that an inducing-stretch cycle $G[x,y]$ is *$i$–minimum* when $G[x,y]$ has exactly $3i - 1$ vertices.

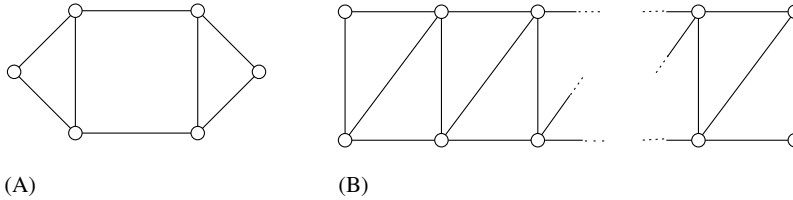

**Figure 10** (A) A graph with stretch number 3/2. Since it is prime and it has more than 5 vertices, then it does not belong to $Gen(*; P_3, C_3, C_5)$. (B) Same properties as in (A) but with arbitrary size.

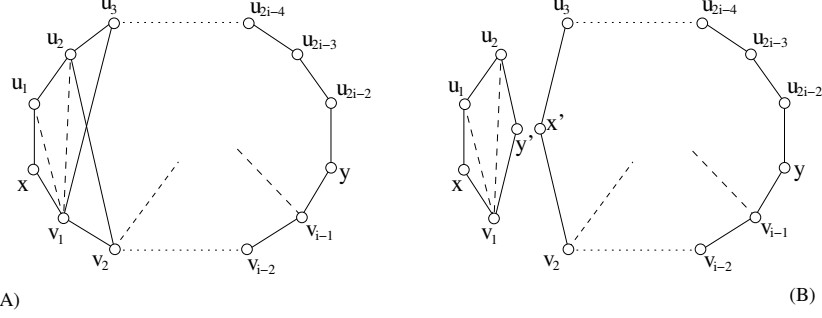

**Figure 11** (A) The cycle $G[x,y]$ described in Lemma 8. Dotted lines represent paths, dashed lines represent chords that may or may not exist. (B) The two components obtained from $G[x,y]$ after the split.

Both Lemmas 8 and 9 require the following claim:

**Claim 1** *Let $G \in \text{sDH}(2)$. Then, $G$ does not contain a cycle $C_n$, with $n \geq 6$ and $cd(C_n) \leq 1$, as induced subgraph.*

**Lemma 8** *Let $i \geq 2$ be an integer, and let $G$ be a graph such that $s(G) = 2 - \frac{1}{i}$. If $G[x,y]$ is a $i$–minimum inducing-stretch cycle of $G$, then $G[x,y]$ is split decomposable into a sequence of $C_5$ cycles.*

**Proof** According to the hypotheses, we may assume that $P_G(x,y) = (x, u_1, u_2, \ldots, u_{2i-2}, y)$ and $p_G(x,y) = (x, v_1, v_2, \ldots, v_{i-1}, y)$. The subgraph $G[x,y]$ induced by $P_G(x,y)$ and $p_G(x,y)$ is shown in Fig. 11A. If $i = 2$, then $G[x,y]$ contains 5 vertices. Since $G[x,y]$ coincides with a cycle $C_5$ the lemma is proven. In the remainder of the proof we assume $i \geq 3$. According to Lemma 2 and to notation about leftmost and rightmost chords of cycles, it follows that $r_j$ and $l_j$ are defined for each $1 \leq j \leq i-1$. We first prove that each one of the following properties about chords hold:

1. $r_1 = 3$;
2. $l_2 = 2$;
3. $l_j \geq 3$ for each $3 \leq j \leq i-1$.

In this way, we prove that $G[x,y]$ contains a split as depicted in Fig. 11A. The components obtained by that split form a cycle $C'$ (isomorphic to a cycle $C_5$) and a cycle $C''$ - see Fig. 11B. Then, we complete the proof by showing that the statement of the lemma can be recursively applied to $C''$.

1. If $r_1 > 3$ then the cycle induced by $v_1, x, u_1, u_2, \ldots, u_{r_1}$ has at least 6 vertices and chord distance at most one. According to Claim 1, this is a contradiction. If $r_1 < 3$ then the induced path $(v_1, u_{r_1}, u_{r_1+1}, \ldots, u_{2i-2}, y)$ and the path $(v_1, v_2, \ldots, v_{i-1}, y)$ provide the following lower bound on $s_G(v_1, y)$:
$$s_G(v_1, y) \geq \frac{2i-2}{i-1} = 2$$
This contradicts $s(G) = 2 - \frac{1}{i}$. Hence, $r_1 = 3$.

2. If $l_2 > 2$ then the cycle induced by $v_2, v_1, x, u_1, u_2, \ldots, u_{l_2}$ has at least 6 vertices and chord distance at most one. According to Claim 1, this is a contradiction. If $l_2 < 2$ then $l_2 = 1$. The induced path $(u_1, u_2, \ldots, u_{2i-2}, y)$ and the path $(u_1, v_2, v_3, \ldots, v_{i-1}, y)$ provide the following lower bound on $s_G(u_1, y)$:
$$s_G(u_1, y) \geq \frac{2i-2}{i-1} = 2$$
This contradicts $s(G) = 2 - \frac{1}{i}$. Hence, $l_2 = 2$.

3. Assume $l_j < 3$ for some $j$ such that $3 \leq j \leq i-1$. In this case, the induced path $(u_{l_j}, u_{l_j+1}, \ldots, u_{2i-2}, y)$ and the path $(u_{l_j}, v_j, v_{j+1}, \ldots, v_{i-1}, y)$ provide the following lower bound on $s_G(v_{l_j}, y)$:
$$s_G(u_{l_j}, y) \geq \frac{2i-1-l_j}{i-j+1} \geq \frac{2i-3}{i-2}.$$
This contradicts $s(G) = 2 - \frac{1}{i}$. Hence, $l_j \geq 3$, for each $3 \leq j \leq i-1$.

To complete the proof, let us analyze the two components obtained after the split. One component $C'$ is a cycle $C_5$, while the other component is a cycle $C''$ with $3i-4$ vertices. Since $C''$ has $3i-4$ vertices, and since the minimum number of vertices to get the stretch $2 - \frac{1}{i}$ is $3i-1$, then $s(C'') < 2 - \frac{1}{i}$. Moreover, the induced paths $(x', u_3, u_4, \ldots, u_{2i-2}, y)$ and $(x', v_2, v_3, \ldots, v_{i-1}, y)$ provide the following lower bound on $s_{C''}(x', y)$:
$$s_{C''}(x', y) \geq \frac{2i-3}{i-1} = 2 - \frac{1}{i-1}.$$

As a consequence, $s(C'') = s_{C''}(x', y) = 2 - \frac{1}{i-1}$.

Hence, $C''$ coincides with $C''[x', y]$ and it is a $(i-1)$–minimum inducing-stretch cycle. Finally, since $i \geq 3$, the arguments above can be recursively applied to $C''$ by replacing $i$ with $i-1$. It follows that $G[x, y]$ can be splitted into a sequence of $i$ cycles $C_5$. Notice that this implies $G[x, y] \in Gen(*; P_3, C_3, C_5)$. $\square$

**Lemma 9** *Let $i \geq 2$ be an integer, and let $G$ be a graph such that $s(G) = 2 - \frac{1}{i}$. If $G[x, y]$ is not a $i$–minimum inducing-stretch cycle of $G$, then*

- *$G[x, y]$ is split decomposable into a sequence of $C_5$ and $C_4$ cycles;*
- *$G[x, y]$ contains an $i$–minimum inducing-stretch cycle of $G$ as induced subgraph.*

**Proof** According to hypotheses, we can assume that $P_G(x, y) = (x, u_1, u_2, \ldots, u_{(2i-1)s-1}, y)$ and $p_G(x, y) = (x, v_1, v_2, \ldots, v_{is-1}, y)$, for some integer $s \geq 2$. The subgraph $G[x, y]$ induced by $P_G(x, y)$ and $p_G(x, y)$ is shown in Fig. 12. According to Lemma 2 and to notation about leftmost and rightmost chords of cycles, it follows that $r_j$ and $l_j$ are defined for each $1 \leq j \leq is-1$. We first prove that each one of the following properties about chords hold:

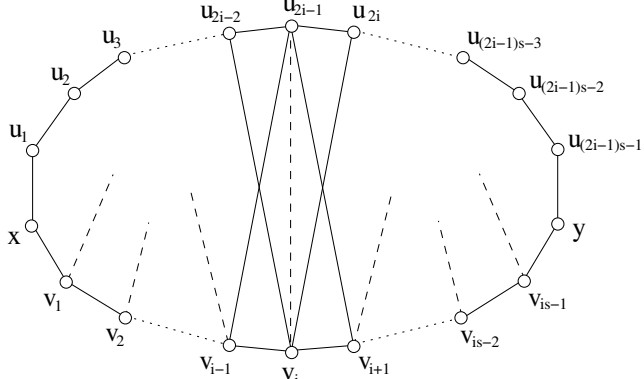

**Figure 12  The cycle $G[x,y]$ described in Lemma 9.** Dotted lines represent paths, dashed lines represent chords that may or may not exist.

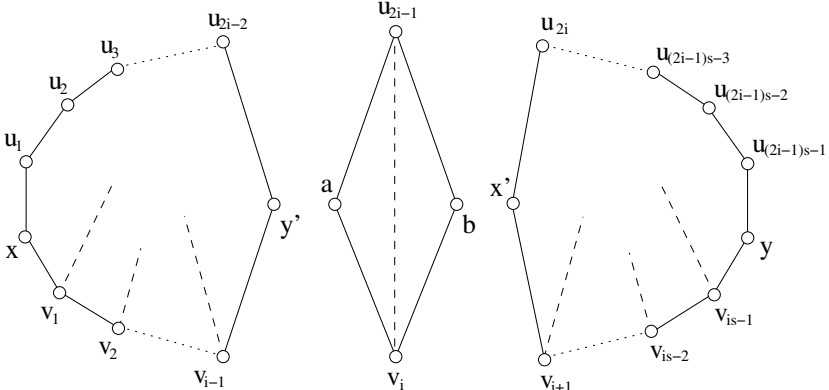

**Figure 13  The three components obtained from the cycle $G[x,y]$ shown in Fig. 12.**

1. $r_{i-1} = 2i - 1$
2. $r_j \leq 2i - 2$, for each $1 \leq j \leq i - 2$
3. $r_i = 2i$
4. $l_i = 2i - 2$
5. $l_{i+1} = 2i - 1$
6. $l_j \geq 2i$, for each $i + 2 \leq j \leq is - 1$

In this way, we prove that $G[x,y]$ contains two splits as depicted in Fig. 12. The components obtained by that split form a cycle $C'$, a cycle $C''$ (isomorphic to a cycle $C_4$) and a cycle $C'''$ (see Fig. 13). Then, we complete the proof by showing that Lemma 8 can be applied to $C'$, and that the current lemma can be recursively applied to $C'''$.

1. If $r_{i-1} > 2i - 1$ then the induced path $(x, u_1, u_2, \ldots, u_{r_{i-1}})$ and the path $(x, v_1, v_2, \ldots, v_{i-1}, u_{r_{i-1}})$ provide the following lower bound on $s_G(u_1, u_{r_{i-1}})$:

$$s_G(u_1, u_{r_{i-1}}) \geq \frac{r_{i-1}}{i} > \frac{2i - 1}{i} = 2 - \frac{1}{i}.$$

This contradicts $s(G) = 2 - \frac{1}{i}$. Conversely, if $r_{i-1} < 2i - 1$ then the induced paths $(v_{i-1}, u_{r_{i-1}}, u_{r_{i-1}+1}, \ldots, u_{(2i-1)s-1}, y)$ and $(v_{i-1}, v_i, \ldots, v_{is-1}, y)$ provide the following lower bound on $s_G(v_{i-1}, y)$:

$$s_G(v_{i-1}, y) \geq \frac{(2i-1)s - r_{i-1} + 1}{is - (i-1)} > \frac{(2i-1)s - (2i-1) + 1}{is - i + 1} \geq \frac{(2i-1)s - (2i-3)}{is - i + 1}.$$

This contradicts $s(G) = 2 - \frac{1}{i}$, because it can easily observed that $\frac{(2i-1)s-(2i-3)}{is-i+1} > 2 - \frac{1}{i}$ holds for each $i$ and for each $s$. Hence, $r_{i-1} = 2i - 1$.

2. Assume $r_j > 2i - 2$, for some $j$ such that $1 \leq j \leq i - 2$. In this case, the induced path $(x, u_1, u_2, \ldots, u_{r_j})$ and the path $(x, v_1, v_2, \ldots, v_j, u_{r_j})$ provide the following lower bound on $s_G(x, u_{r_j})$:

$$s_G(x, u_{r_j}) \geq \frac{r_j}{j+1} > \frac{2i-2}{i-1} = 2$$

This contradicts $s(G) = 2 - \frac{1}{i}$.

3. If $r_i < 2i$ then the induced paths $(v_i, u_{r_i}, u_{r_i+1}, \ldots, u_{(2i-1)s-1}, y)$ and $(v_i, v_{i+1}, \ldots, v_{is-1}, y)$ provide the following lower bound on $s_G(v_i, y)$:

$$s_G(v_i, y) \geq \frac{(2i-1)s - r_i + 1}{is - i} > \frac{(2i-1)s - 2i + 1}{is - i} = 2 - \frac{1}{i}$$

This contradicts $s(G) = 2 - \frac{1}{i}$. Conversely, if $r_i > 2i$ then consider the induced path $(x, u_1, u_2, \ldots, u_{r_i})$ and the path $(x, v_1, v_2, \ldots, v_i, u_{r_i})$. The length of the first path is $r_i > 2i \geq 2i + 1$, while the length of the second one is $i + 1$. These values provide the following lower bound on $s_G(x, u_{r_i})$:

$$s_G(x, u_{r_i}) \geq \frac{2i+1}{i+1} = 2 - \frac{1}{i+1} > 2 - \frac{1}{i}$$

This contradicts $s(G) = 2 - \frac{1}{i}$. Hence, $r_i = 2i$.

4. If $l_i < 2i - 2$ then consider the induced paths $(u_{l_i}, u_{l_i+1}, \ldots, u_{(2i-1)s-1}, y)$ and $(u_{l_i}, v_i, v_{i+1} \ldots, v_{is-1}, y)$. The length of the first path is $(2i-1)s - l_i > (2i-1)s - (2i-2) \geq (2i-1)s - (2i-2) + 1$, while the length of the second one is $is - i + 1$. These values provide the following lower bound on $s_G(u_{l_i}, y)$:

$$s_G(u_{l_i}, y) \geq \frac{(2i-1)s - (2i-2) + 1}{is - i + 1} = 2 - \frac{s-1}{is - i + 1}$$

This contradicts $s(G) = 2 - \frac{1}{i}$, because $2 - \frac{s-1}{is-i+1} > 2 - \frac{1}{i}$ is equivalent to $1 > 0$. Conversely, if $l_i > 2i - 2$ then the induced paths $(x, u_1, u_2, \ldots, u_{l_i}, v_i)$ and $(x, v_1, v_2, \ldots, v_i)$ provide the following lower bound on $s_G(x, u_{r_i})$:

$$s_G(x, v_i) \geq \frac{l_i + 1}{i} > \frac{2i - 2 + 1}{i} = 2 - \frac{1}{i}$$

This contradicts $s(G) = 2 - \frac{1}{i}$. Hence, $l_i = 2i - 2$.

5. If $l_{i+1} < 2i - 1$ then consider the induced path $(u_{l_{i+1}}, u_{l_{i+1}+1}, \ldots, u_{(2i-1)s-1}, y)$ and the path $(u_{l_{i+1}}, v_{i+1}, v_{i+2}, \ldots, v_{is-1}, y)$. The length of the first path is $(2i-1)s - l_{i+1} > (2i-1)s - (2i-1)$, while the length of the second one is $is - (i+1) + 1$. These values provide the following lower bound on $s_G(u_{l_{i+1}}, y)$:

$$s_G(u_{l_{i+1}}, y) > \frac{(2i-1)s - (2i-1) + 1}{is - i} = 2 - \frac{1}{i}.$$

This stretch contradicts $s(G) = 2 - \frac{1}{i}$. Conversely, if $l_{i+1} > 2i - 1$ then consider the induced paths $(x, u_1, u_2, \ldots, u_{l_{i+1}}, v_{i+1})$ and $(x, v_1, v_2, \ldots, v_i, v_{i+1})$. The length of the first path is $l_{i+1} + 1 > (2i-1) + 1 \geq 2i + 1$, while the length of the second one is $i + 1$. These

values provide the following lower bound on $s_G(x, v_{i+1})$:

$$s_G(x, v_{i+1}) \geq \frac{2i+1}{i+1} = 2 - \frac{1}{i+1} > 2 - \frac{1}{i}$$

This contradicts $s(G) = 2 - \frac{1}{i}$. Hence, $l_{i+1} = 2i - 1$.

6. Assume $l_j < 2i$, for some $j$ such that $i + 2 \leq j \leq is - 1$. In this case, the induced path $(u_{l_j}, u_{l_j+1}, \ldots, u_{(2i-1)s-1}, y)$ and the path $(u_{l_j}, v_j, v_{j+1}, \ldots, v_{is-1}, y)$ provide the following lower bound on $s_G(u_{l_j}, y)$:

$$s_G(u_{l_j}, y) \geq \frac{(2i-1)s - l_j}{is - j + 1} > \frac{(2i-i)s - 2i}{is - (i+2) + 1} = 2 - \frac{s-2}{is - i - 1}$$

This contradicts $s(G) = 2 - \frac{1}{i}$, because $2 - \frac{s-2}{is-i-1} > 2 - \frac{1}{i}$ is equivalent to $i > 1$. The latter inequality holds because $i \geq 2$ by hypothesis.

To complete the proof, let us analyze the three components obtained after the split.

- The first component $C'$ is a cycle with $3i - 1$ vertices. The induced paths $(x, u_1, u_2, \ldots, u_{2i-2}, y')$ and $(x, v_1, v_2, \ldots, v_{i-1}, y')$ provide the following lower bound on $s_{C'}(x, y')$:

$$s_{C'}(x, y') \geq \frac{2i-1}{i} = 2 - \frac{1}{i}$$

Since $s(G) = 2 - \frac{1}{i}$ and $C'$ is an induced subgraph of $G$, then $s(C') = s_{C'}(x, y') = 2 - \frac{1}{i}$ and $C'$ coincides with $C'[x, y']$. Hence, $C'$ is a $i$–minimum inducing-stretch cycle which the statement of Lemma 8 can be applied to. According to such a Lemma, $C'$ can be splitted into a sequence of cycles $C_5$.

- The second component is a cycle $C''$ isomorphic to a cycle $C_4$. Notice that $C''$ can be splitted into two cycles $C_3$.

- This component is a cycle $C'''$ with $(is - 1 - i) + 2 + [(2i-1)s - 1 - 2i + 1] = (3i-1)(s-1)$ vertices. The induced paths $(x', u_{2i}, u_{2i+1}, \ldots, u_{(2i-1)s-1}, y)$ and $(x', v_{i+1}, v_{i+2}, \ldots, v_{is-1}, y)$ provide the following lower bound on $s_{C'''}(x', y)$:

$$s_{C'''}(x', y) \geq \frac{(2i-1)s - 2i + 1}{is - i} = \frac{(2i-1)(s-1)}{is - i} = 2 - \frac{1}{i}$$

Since $s(G) = 2 - \frac{1}{i}$ and $C'''$ is an induced subgraph of $G$, then $s(C''') = s_{C'''}(x', y) = 2 - \frac{1}{i}$ and $C'''$ coincides with $C'''[x', y]$. Now, if $s = 2$ then $C'''$ is a $i$–minimum inducing-stretch cycle and we can recursively apply Lemma 8 to it. If $s > 2$ then $C'''$ is not a $i$–minimum inducing-stretch cycle, and we can apply this lemma to it. Hence, according to the analysis of cycles $C'$ and $C''$, $C'''$ can be splitted into a sequence of cycles $C_4$ and $C_5$.

Summarizing, it follows that $G[x, y]$ is split decomposable into a sequence of $C_5$ and $C_4$ cycles, and it contains an $i$–minimum inducing-stretch cycle of $G$ as induced subgraph. Moreover, it follows that $G[x, y] \in Gen(*; P_3, C_3, C_5)$. □

The above lemmata induce the following additional results.

**Corollary 3** *Let $G$ be a graph such that $s(G) = 2 - 1/i$, for some $i \geq 2$. There exists a cycle-pair $\{u, v\}$ in $G$ such that the induced subgraph $G[u, v]$ can be decomposed into a sequence of $i - 1$ cycles $C_5$.*

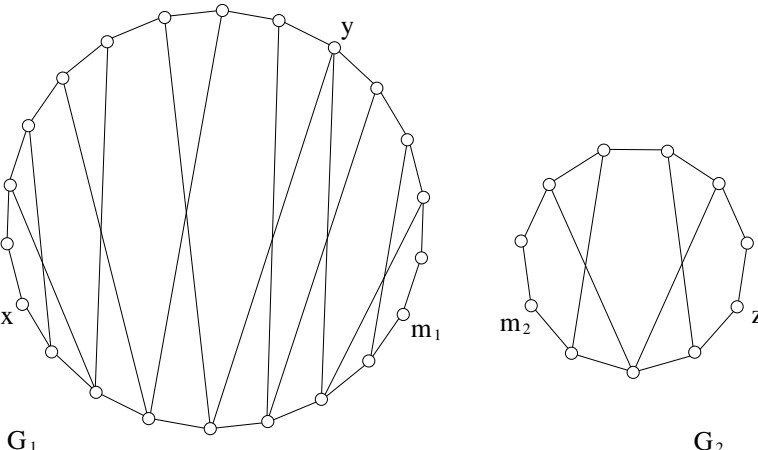

**Figure 14** **Two graphs $G_1$ and $G_2$ that can be used to show that Theorem 6 in [*11*] is not correct.**

**Corollary 4** *Let  $G$ be a graph such that  $s(G) = 2 - 1/i$, for some  $i \geq 2$. For each cycle-pair $\{u,v\}$ such that  $s_G(x,y) = s(G)$, the induced subgraph  $G[u,v]$ belongs to  $Gen(*; P_3, C_3, C_5)$.*

## COMPUTING THE STRETCH NUMBER

In *Cicerone & Di Stefano (2001)* it has been shown that computing the stretch number for arbitrary graphs is NP-hard. In *Cicerone (2011b)* it is stated that in $Gen(*; P_3, C_3, C_5)$ it can be computed in linear time. This result was based on the following property (cf. Theorem 6 in *Cicerone (2011b)*):

- Given $G = (V, E)$, let $s_G(v) = \max\{s_G(v,u) | u \in V\}$, and denote as $\overline{v}$ any vertex such that $s_G(v) = s_G(v, \overline{v})$.
- Let  $G_1 = (V_1, E_1)$ and  $G_2 = (V_2, E_2)$ be two graphs. Let  $m_1 \in V_1$ such that  $s_{G_1}(m_1) = 2 - 1/i$, and let  $m_2 \in V_2$ such that  $s_{G_2}(m_2) = 2 - 1/j$. If  $G = (G_1, m_1) * (G_2, m_2)$, then

  $s(G) = \max\{s(G_1), s(G_2), s_G(\overline{m_1}, \overline{m_2})\},$

  *where*  $s_G(\overline{m_1}, \overline{m_2}) = 2 - \frac{1}{i+j-1}$.

Unfortunately, the following counterexample shows that this property is not true. Considering the graphs $G_1$ and $G_2$ as represented in Fig. 14. It can be observed that both the graphs can be obtained by composing holes with five and four vertices, namely $G_1 = H_5 * H_5 * H_5 * H_5 * H_4 * H_5 * H_5$ and $G_2 = H_5 * H_5 * H_5$. Concerning their stretch numbers, we get $s(G_1) = s_{G_1}(x,y) = \frac{9}{5} = 2 - \frac{1}{5}$ and $s(G_2) = s_{G_2}(m_2,z) = \frac{7}{4} = 2 - \frac{1}{4}$.

By focusing on $m_1$ and $m_2$, we get that $\overline{m_1} \equiv x$ and $s_{G_1}(m_1) = s_{G_1}(m_1, x) = \frac{14}{8} = \frac{7}{4}$, whereas $\overline{m_2} \equiv z$ and $s_{G_2}(m_2) = s_{G_2}(m_2,z) = 2 - \frac{1}{4}$. Hence, by referring to the notation used in the above property we have $i = j = 4$.

Consider now the composed graph $G = (G_1, m_1) * (G_2, m_2)$. According to the property above, we get $s(G) = \max\{s(G_1), s(G_2), s_G(\overline{m_1}, \overline{m_2})\} = \max\{s_{G_1}(x,y), s_{G_2}(m_2,z), s_G(x,z)\} =$

$\max\{\frac{9}{5}, \frac{7}{4}, \frac{20}{11}\} = \frac{20}{11}$. Unfortunately, Theorem 1 implies that $\frac{20}{11}$ is not an admissible stretch number. Additionally, the above property further states that $s_G(\overline{m_1}, \overline{m_2}) = 2 - \frac{1}{i+j-1}$, but in the example $s_G(\overline{m_1}, \overline{m_2}) = s_G(x, z) = \frac{20}{11}$ whereas $2 - \frac{1}{i+j-1} = \frac{13}{7}$. In reality, it can been easily observed that $s(G) = s_G(y, m_2) = \frac{11}{6}$. Hence, the above property does not hold.

In the remainder of this section we provide a correct linear time algorithm for computing the stretch number in the class $Gen(*; P_3, C_3, C_5)$. The technique underlying the algorithm is the following:

1. decompose $G$ by using a split decomposition algorithm;
2. compute the stretch number of all the components in $\mathcal{D}(G)$ (by exploiting Theorem 6);
3. starting from the components in $\mathcal{D}(G)$, rebuild $G$ by exploiting the split composition: at each step use some *relationship* that allow us to compute $s(G)$ from $s(G_1)$ and $s(G_2)$ when $G = G_1 * G_2$.

It is clear that such an approach works only if the relationship at Step 3 exists: fortunately, in the subsequent Theorem 8 we show such a relationship for graphs in the class sDH(2). Also, we will show that for graphs in $Gen(*; P_3, C_3, C_5)$, steps 2 and 3 can be performed in linear time. Before presenting the main theorem and the subsequent algorithm, we provide some properties for graphs in sDH(2).

### Notation and properties for graphs in sDH(2)

Given a pair $(u, v)$ of distinct vertices in any graph $G \in$ sDH(2), we are interested not only in the stretch number $s_G(u, v)$ they produce, but also to the distances that generated such stretch. To this end we use the symbol $\delta_G(\cdot)$ to represent the *distance-pair* between the vertices $u$ and $v$ in $G$, that is:

$$\delta_G(u, v) = (D_G(u, v), d_G(u, v))$$

According to Lemma 3, if $G = (G_1, m_1) * (G_2, m_2)$, $u \in G_1 \setminus N_{G_1}[m_1]$, and $v \in G_2 \setminus N_{G_2}[m_2]$, then we get

$$\delta_G(u, v) = (D_{G_1}(u, m_1) + D_{G_2}(v, m_2) - 1, d_{G_1}(u, m_1) + d_{G_2}(v, m_2) - 1).$$

However, if $\delta_{G_1}(u, m_1) = (p_1, q_1)$ and $\delta_{G_2}(u, m_2) = (p_2, q_2)$, it is difficult to represent

$$\delta_G(u, v) = (p_1 + p_2 - 1, q_1 + q_2 - 1) \tag{1}$$

in terms of $\delta_{G_1}(u, m_1)$ and $\delta_{G_2}(u, m_2)$ and then to compute $s_G(u, v)$. In order to simplify the computation of $s_G(u, v)$, we prefer to encode the distances between $u$ and $v$ with a different pair of natural numbers, called *distance-copair*. The function $\eta(\cdot)$ transforms a distance-pair $(p, q)$ into a distance-copair. By definition:

$$\eta((p, q)) = \langle p - q, 2q - p - 1 \rangle.$$

Note that distance-copairs are represented by angle brackets to easily distinguish them from distance-pairs. It is not difficult to prove that $\eta$ is bijective and that

$$\eta^{-1}(\langle a, b \rangle) = (2a + b + 1, a + b + 1).$$

Then, given two vertices $u, v$ of a graph $G$, as $\delta_G(u, v)$ provides the distance-pair between $u, v$, the next function $\zeta_G(\cdot, \cdot)$ provides the distance-copair between the same pair of vertices:

$$\zeta_G(u, v) = \eta(\delta_G(u, v)).$$

Now, when $G = (G_1, m_1) * (G_2, m_2)$ and $u$ and $v$ are such that $u \in G_1 \setminus N_{G_1}[m_1]$ and $v \in G_2 \setminus N_{G_2}[m_2]$, we can compute $\zeta_G(u, v)$ starting from $\zeta_{G_1}(u, m_1)$ and $\zeta_{G_2}(v, m_2)$:

$$\zeta_G(u, v) = \zeta_{G_1}(u, m_1) + \zeta_{G_2}(v, m_2) \tag{2}$$

where the $+$ operator used in the above equation is such that:

$$\langle a_1, b_1 \rangle + \langle a_2, b_2 \rangle = \langle a_1 + a_2, b_1 + b_2 \rangle.$$

We can prove that Eq. (2) holds by calculating $\delta_G(u, v)$ when $\delta_{G_1}(u, m_1) = (p_1, q_1)$ and $\delta_{G_2}(u, m_2) = (p_2, q_2)$, as done in Eq. (1). For the left side of Eq. (2), by applying the $\eta^{-1}$ function, we have:

$$\eta^{-1}(\zeta_G(u, v)) = \delta_G(u, v)$$

For the right side:

$$
\begin{aligned}
\eta^{-1}(\zeta_{G_1}(u, m_1) + \zeta_{G_2}(v, m_2)) &= \eta^{-1}(\langle p_1 - q_1, 2q_1 - p_1 - 1 \rangle + \langle p_2 - q_2, 2q_2 - p_2 - 1 \rangle) \\
&= \eta^{-1}(\langle p_1 - q_1 + p_2 - q_2, 2q_1 + 2q_2 - p_1 - p_2 - 2 \rangle) \\
&= (p_1 + p_2 - 1, q_1 + q_2 - 1) \\
&= \delta_G(u, v).
\end{aligned}
$$

Then Eq. (2) holds being $\eta$ bijective. For sake of simplicity, we remove the subscript $G$ from function $\zeta$ whenever the graph under consideration is clear from the context.

Given any distance-copair $\langle a, b \rangle$,

$$\sigma(\langle a, b \rangle) = \frac{2a + b + 1}{a + b + 1},$$

that is $\sigma(\langle a, b \rangle)$ represents the stretch number produced by the distance-pair $\eta^{-1}(\langle a, b \rangle)$. In particular, notice that if $u$ and $v$ are two distinct vertices of $G$ then

$$s(G[u, v]) = \sigma(\zeta_G(u, v)) \tag{3}$$

Again, for sake of simplicity, we simply write $\sigma(a, b)$ instead of $\sigma(\langle a, b \rangle)$. According to Theorem 1, we know that if $s(G) < 2$, then $s(G) = 2 - 1/i$, for some integer $i \geq 1$. It is worth to remark that $\sigma(i, 0)$, with $i \geq 0$, gives exactly all the possible stretch numbers smaller than two. In fact:

$$\sigma(0, 0) = \frac{1}{1}, \sigma(1, 0) = \frac{3}{2}, \sigma(2, 0) = \frac{5}{3}, \sigma(3, 0) = \frac{7}{4}, \sigma(4, 0) = \frac{9}{5}, \dots, \sigma(i, 0) = 2 - \frac{1}{i+1}, \dots$$

There is an interesting interpretation of a distance-copair $\langle n, m \rangle$. Indeed, it can be thought as the distance-copair of two vertices $x, y$ with maximum stretch number in any graph $G$ obtained by the split composition of $n$ cycles $C_5$ and $m$ cycles $C_4$, when $T(G)$ is a path and the marked vertices of a component are at distance two. This is the case of the

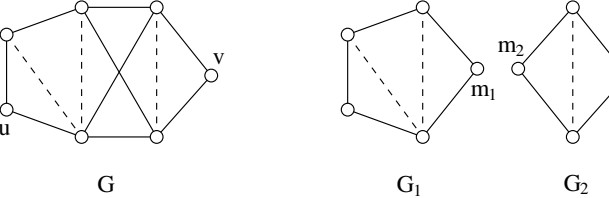

**Figure 15** **Dashed edges represent chords that may or may not exist.** $(u, v)$ is a stretch-pair in $G$, $\delta_G(u, v) = (4, 3)$ and $\zeta_G(u, v) = \langle 1, 1 \rangle$. Notice that the distance-copair $\langle 1, 1 \rangle$ represents exactly the number of cycles $C_4$ and $C_5$ to be joined by the split composition in order to get $G$ (cf. cycles $G_1$ and $G_2$ to be joined via $m_1$ and $m_2$ to get $G$).

graphs $G[x, y]$ mentioned in the statements of Lemmas 8 and 9 (see also Figs. 11, 12 and 13). The case with a graph $G$ obtained by the split composition of a $C_5$ cycle and a $C_4$ cycle is given in Fig. 15. Here the distance-copair of vertices $u, v$ is $\langle 1, 1 \rangle$, since $\zeta_{G_1}(u, m_1) = \langle 1, 0 \rangle$ and $\zeta_{G_2}(v, m_2) = \langle 0, 1 \rangle$. Note that $G[u, v]$ is not an inducing stretch cycle of $G$, being $\sigma(\langle 1, 1 \rangle) = 4/3$, whereas $s(G) = 3/2$.

Regarding the sum operator for distance-copairs, note that it is both commutative and associative, then we write:

$$k\langle a, b \rangle = \underbrace{\langle a, b \rangle + \langle a, b \rangle + \cdots + \langle a, b \rangle}_{k \text{ times}}$$

and then, in particular, $a\langle 1, 0 \rangle = \langle a, 0 \rangle$.

We also introduce a partial order in the set of the distance-copairs:

- $\langle x, b \rangle < \langle y, b \rangle$ for $x < y$ and each $b$;
- $\langle a, x \rangle < \langle a, y \rangle$ for $x > y$ and each $a \neq 0$.

As a consequence, given a sets of comparable distance-copairs, it is possible to find the maximum distance-copair, as we will see in Algorithm 1.

Regarding the $\sigma$ function, we have the following properties:

- $\langle x, b \rangle < \langle y, b \rangle \iff \sigma(x, b) < \sigma(y, b)$ for each $b$;
- $\langle a, x \rangle < \langle a, y \rangle \iff \sigma(a, x) < \sigma(a, y)$ for each $a \neq 0$.

Then the stretch number $\sigma(a, x)$ decreases when $x$ increases. Moreover, when $a = 0$, $\sigma(a, x) = 1$ for each possible value of $x$.

A further property of the $\sigma$ function is that, for each integer $k > 0$, we get

$$\sigma(ka, k - 1) = \sigma(a, 0), \tag{4}$$

and, in particular, $\sigma(2a, 1) = \sigma(a, 0)$.

Given the set $\mathcal{S}(G) = \{(u, v) : \delta_G(u, v) = (2i + 1, i + 1) \text{ for some integer } i \geq 0\}$, we call *stretch-pair* of $G$ each element in $\mathcal{S}$. It follows that if $(u, v) \in \mathcal{S}(G)$, then $\zeta_G(u, v) = \langle i, 0 \rangle$. Among the elements of $S$ there are pairs of vertices that define the $(i + 1)$–minimum inducing-stretch cycles of $G$. Note that not all the pairs of vertices are stretch-pairs: an example is given by the pairs of vertices at distance two in a chordless cycle $C_4$. Anyway, it

can be observed that given any vertex $v$, there exists at least one stretch-pair involving $v$. In fact, if $v$ and $v'$ are adjacent in $G$, then $(v, v') \in \mathcal{S}(G)$ and $\zeta(v, v') = \langle 0, 0 \rangle$. Given any vertex $v$, we denote $\tau_G(v)$ (or just $\tau(v)$, when $G$ is clear by the context) any vertex of $G$ such that both the following properties hold:

- $(v, \tau_G(v))$ is a stretch-pair in $\mathcal{S}(G)$;
- $\sigma(\zeta_G(v, \tau_G(v)))$ is maximum.

Moreover with $\zeta_G(v)$ we denote $\zeta_G(v, \tau(v))$, and by $\zeta(G)$ we denote $\zeta_G(v)$ for a vertex $v$ such that $\sigma(\zeta_G(v, \tau_G(v)))$ is maximum. Notice that $\zeta(G)$ corresponds to the distance-copair $\langle i, 0 \rangle$ where $i$ is such that $s(G) = \sigma(i, 0)$.

The following theorem exploits all the above notation to provide a property useful to compute $s(G)$ in any graph $G$ obtained by composing two graphs in sDH(2) via split composition.

**Theorem 8** *Let $G_1 = (V_1, E_1)$ and $G_2 = (V_2, E_2)$ be two graphs in sDH(2), let $m_1 \in V_1$, and let $m_2 \in V_2$. If $G = (G_1, m_1) * (G_2, m_2)$, then*

$$s(G) = \max\{s(G_1), s(G_2), \sigma(\zeta_{G_1}(m_1) + \zeta_{G_2}(m_2))\}.$$

**Proof** According to Lemma 4, it is sufficient to assume that $s(G) > \max\{s(G_1), s(G_2)\}$ and then to prove that if $s(G) = s_G(u, v)$ for some $u \in V_1 \setminus N_{G_1}[m_1]$ and $v \in V_2 \setminus N_{G_2}[m_2]$, then $s(G) = s_G(u, v) = \sigma(\zeta_{G_1}(m_1) + \zeta_{G_2}(m_2))$ - see Fig. 16 for an example. By construction,

$$
\begin{aligned}
s(G) &= s_G(u, v) \\
&\geq s(G[\tau_{G_1}(m_1), \tau_{G_2}(m_2)]) \\
&= s((G_1[\tau_{G_1}(m_1), m_1], m_1) * (G_2[m_2, \tau_{G_2}(m_2)], m_2)) \\
&= \sigma(\zeta_{G_1}(m_1) + \zeta_{G_2}(m_2)).
\end{aligned}
$$

where the last equality is obtained by applying Eqs. (2) and (3).

In what follows we show that $s(G) = s_G(u, v) \leq \sigma(\zeta_{G_1}(m_1) + \zeta_{G_2}(m_2))$.

Since $u \in V_1 \setminus N_{G_1}[m_1]$ and $v \in V_2 \setminus N_{G_2}[m_2]$, the shortest path from $u$ to $v$ must pass through a vertex $x_1$ of $V_1$ in $N_{G_1}(m_1)$ and a vertex $x_2$ of $V_2$ in $N_{G_2}(m_2)$. The same for the longest path from $u$ to $v$ that must pass through a vertex $y_1$ of $V_1$ in $N_{G_1}(m_1)$ and a vertex $y_2$ of $V_2$ in $N_{G_2}(m_2)$. As $G$ is not distance-hereditary (otherwise, both $u$ and $v$ are in $V_1$ or $V_2$, against the hypothesis), these four vertices are all distinct. Moreover each of these paths can not include more than two vertices in $N_{G_1}(m_1) \cup N_{G_2}(m_2)$ otherwise they would be not induced.

Since both $G_1$ and $G_2$ belong to sDH(2), by Corollary 3 we can assume $s(G_1) = \sigma(a_1, 0)$ and $s(G_2) = \sigma(a_2, 0)$ for some integers $a_1, a_2 \geq 0$. Without loss of generality, let $a_1 \geq a_2$. Moreover, by Lemma 7 we get that also $s(G)$ belongs to sDH(2), and hence by Lemma 9 $G[u, v]$ can be obtained by a sequence of split compositions of $C_5$ and $C_4$ cycles. Notice that both $x_1$ and $y_1$ belongs to a component of such a decomposition, as well as $x_2$ and $y_2$.

If $x_1$ and $y_1$ (or $x_2$ and $y_2$) belongs to a $C_4$ cycle, the maximum number of consecutive $C_5$ graphs in the decomposition of $G[u, v]$ is bounded by $a_1$. Then:

$$s(G[u, v]) = \sigma(k_1\langle 1, 0 \rangle + k_2\langle 1, 0 \rangle + \cdots + k_l\langle 1, 0 \rangle, h_1\langle 0, 1 \rangle + h_2\langle 0, 1 \rangle + \cdots + h_{l-1}\langle 0, 1 \rangle),$$

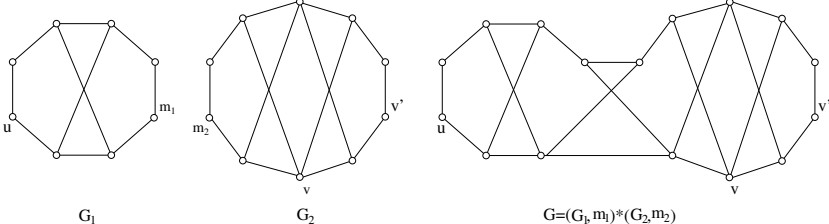

**Figure 16** **An example for Theorem 8.** Here $s(G_1) = s_{G_1}(u, m_1) = 5/3$, $s(G_2) = s_{G_2}(m_2, v) = s_{G_2}(m_2, v') = 3/2$. Notice $(m_2, v)$ is a stretch-pair in $G_2$ whereas $(m_2, v')$ is not. Moreover, $u = \tau_{G_1}(m_1)$ and $v = \tau_{G_2}(m_2)$, and $s(G) = s_G(u, v) = 7/4$ against $s_G(u, v') = 5/3$.

where $k_i$ is the number of $C_5$ in the $i$th sequence of consecutive $C_5$ graphs in the decomposition of $G[u, v]$. Similarly for $h_j$. Note that if the sequences of consecutive $C_5$ graphs are $l$, the sequences of consecutive $C_4$ graphs are $l - 1$ since $u$ and $v$ belongs to $C_5$ components.

Then $s(G[u, v]) = \sigma((\sum_{i=1}^{l} k_i)\langle 1, 0 \rangle + (\sum_{j=1}^{l-1} h_j)\langle 0, 1 \rangle)$. Since $k_i \leq a_1$, for each $i$, and $h_j \geq 1$ for each $j$, we have:

$$s(G[u, v]) \leq \sigma((l \cdot a_1)\langle 1, 0 \rangle + (l - 1)\langle 0, 1 \rangle) = \sigma(\langle l \cdot a_1, l - 1 \rangle) = \sigma(a_1, 0) = s(G_1),$$

a contradiction.

Then $x_1$ and $y_1$ belong to a $C_5$ cycle in a sequence of consecutive $C_5$ graphs. Let $F$ the graph induced by this sequence. Then $s(F) \leq \sigma(\zeta_{G_1}(m_1) + \zeta_{G_2}(m_2))$, as otherwise, $s(G[m_1, \tau_{G_1}(m_1)]) > \sigma(\zeta_{G_1}(m_1))$ or $s(G[m_2, \tau_{G_2}(m_2)]) > \sigma(\zeta_{G_2}(m_2))$.

Let $\langle c, 0 \rangle = \zeta_{G_1}(m_1) + \zeta_{G_2}(m_2)$. If $\sigma(c, 0) \leq \sigma(a_1, 0)$, then we reach a contradiction as above. Then $c > a_1$ and $k_i \leq c$. As consequence:

$$
\begin{aligned}
s(G) &= s(G[u, v]) \\
&= \sigma((\sum_{i=1}^{l} k_i)\langle 1, 0 \rangle + (\sum_{j=1}^{l-1} h_j)\langle 0, 1 \rangle) \\
&\leq \sigma((l \cdot c)\langle 1, 0 \rangle + (l - 1)\langle 0, 1 \rangle) \\
&\leq \sigma(\langle l \cdot c, l - 1 \rangle) \\
&= \sigma(c, 0) \\
&= \sigma(\zeta_{G_1}(m_1) + \zeta_{G_2}(m_2)).
\end{aligned}
$$

This concludes the proof. □

## The algorithm

Here we assume that $T(G)$ is rooted at a component of $\mathcal{D}(G)$. If $B$ is a vertex of $T(G)$, we denote by $\bar{B}$ the graph obtained by the composition of $B$ and all the descendant vertices of $B$ in $T(G)$.

Given a graph $G$, the following recursive function $Z(\cdot, \cdot)$ computes $\zeta(\bar{B})$ for a generic component $B$ of $T(G)$ and, if $m$ is the marked vertex connecting $B$ to its parent in $T(G)$, it

computes $\zeta_{\bar{B}}(m)$, too. Then the function $Z$ returns the pair $[\zeta(\bar{B}), \zeta_{\bar{B}}(m)]$ and $first(Z(\cdot,\cdot))$ and $second(Z(\cdot,\cdot))$ denote the first and second element of the pair, respectively.

The following algorithm computes $s(G)$ for a graph $G$ such that $s(G) < 2$.

---

**Algorithm 1:** function $Z(T,B)$, $T = T(G)$, for $G \in$ sDH(2)

---

**Input:** $T(G)$ is assumed rooted, $B$ is a component of $T(G)$
**Output:** the pair $[\zeta(\bar{B}), \zeta_{\bar{B}}(m)]$
1   Let $\{B_1, B_2, \ldots, B_k\}$ the children of $B$ connected to $B$ by edges $(m_i, m_i')$, $i = 1, \ldots, k$, where $m_i \in B$
2   Let $m$ the marked vertex connecting $B$ to its parent in $T$, if it exists.
3   Let $a = b = c = d = e = f = \langle 0, 0 \rangle$
4   $a = \max\{\zeta(u,v) \mid (u,v) \text{ is a stretch-pair of } B\}$
5   $b = \max\{first(Z(T,B_i)) \mid i = 1, \ldots, k\}$
6   $c = \max\{\zeta_B(m_i) + second(Z(T,B_i)) \mid i = 1, \ldots, k\}$
7   $d = \max\{\zeta(m_i, m_j) + second(Z(T,B_i)) + second(Z(T,B_j)) \mid (m_i, m_j) \text{ is a stretch-pair of } B\}$
8   **if** $m$ exists **then**
9      $e = \zeta_B(m)$
10     $f = \max\{\zeta(m, m_i) + second(Z(T,B_i)) \mid (m, m_i) \text{ is a stretch-pair of } B\}$
11 **return** $[\max\{a,b,c,d\}, \max\{e,f\}]$

---

---

**Algorithm 2:** computing $s(G)$

---

**Input:** a graph $G$ with any component $B$ of $\mathcal{D}(G)$ such that $s(B) < 2$
**Output:** $s(G)$
1   compute $\mathcal{D}(G)$ and $T(G)$
2   choose a component $B$ of $\mathcal{D}(G)$ as a root of $T(G)$
3   **return** $\sigma(first(Z(T(G), B)))$

---

**Theorem 9** *If $G$ is a graph and any component $B$ of $\mathcal{D}(G)$ is such that $s(B) < 2$ then $s(G)$ can be computed in $O(t(B) \cdot n^2)$ time, where $t(B)$ is the time to compute $\zeta(B)$.*

**Proof** We prove the statement by analyzing Algorithm 2, and assuming that the input graph $G$ has $n$ vertices. By using the algorithm proposed in *Dahlhaus (2000)*, step at Line 1 can be performed in linear time. At Line 2 a component $B$ is chosen as root of $T(G)$, and at the last line the function $Z$ is called on $B$. Then assuming that $first(Z(T(G),B))$ correctly returns $\zeta(\bar{B})$, then $\sigma(\zeta(\bar{B}))$ is $s(G)$, being $G \equiv \bar{B}$. Then we have to analyze Algorithm 1. This algorithm is a recursive algorithm that, working on a node $B$ of $T(G)$ call itself at Lines 5, 6, 7, and 10 for the children of $B$. We assume that these lines are not performed if $B$ has no children or, which is the same, that the function max applied to an empty set returns $\langle 0, 0 \rangle$. Let $u, v$ two vertices of $G$ such that $s(G) = s(G[u,v])$, then there are the following cases:

- $u$ and $v$ are both in $B$, then $\zeta(\bar{B}) = \zeta_B(u,v)$ and this value is stored in variable $a$ at Line 4. It can be computed in time $t(B)$.
- $u$ and $v$ are both in $\bar{B}_i$, for a certain $i$, then this value is stored in variable $b$ at Line 5. It can be computed in $O(k)$ time.
- $u$ is in $B$ and $v$ is in a child of $B$, then by Theorem 8 $u$ is $\tau_B(m_i)$, for a certain $i$, and $\zeta(\bar{B}) = \zeta_B(m_i) + \zeta_{\bar{B}_i}(m_i')$. This value is stored in variable $c$ at Line 6. It can be computed in time $t(B) \cdot k$.
- $u$ and $v$ are in two different children $B_i$ and $B_j$, $i \neq j$, of $B$, then by applying two times Theorem 8, $\zeta(u,v) = \zeta_{\bar{B}_i}(m_i') + \zeta_B(m_i, m_j) + \zeta_{\bar{B}_j}(m_j')$. This value is stored in variable $d$ at Line 7. It can be computed in time $t(B) \cdot k^2$.

The maximum of these four values is returned as $\zeta(\bar{B})$ at Line 11. Note that the distance co-pairs are in general only partially ordered, but the stretch-pairs used in the algorithm are totally ordered and then the maximum function is well defined. It could be possible that $u$ is not in $\bar{B}$, whereas $v$ is. Then, being $Z$ a recursive function it is important to compute and return $\zeta_{\bar{B}}(m)$, since this value could be needed to compute $\zeta(G)$. There are two cases according to $\tau_{\bar{B}}(m)$ is in $B$ or not.

- if $\tau_{\bar{B}}(m)$ is in $B$, then $\zeta_B(m)$ is stored in variable $e$ at Line 9. It can be computed in time $t(B)$.
- if $\tau_{\bar{B}}(m)$ is not in $B$, then by Theorem 8, $\zeta_{\bar{B}}(m) = \zeta_B(m, m_i) + \zeta_{\bar{B}_i}(m_i')$ for a certain $i$ and this value is stored in variable $e$ at Line 9. It can be computed in time $t(B) \cdot k$.

The maximum of these two values is returned as $\zeta(\bar{B})$ at line 11. The overall time for a call of Algorithm 1 is then $O(t(B) \cdot k^2)$. Since it is sufficient a single call of the $\zeta$ function for each component of $\mathcal{D}(G)$ and all the children of the components sum up to $O(n)$ then the overall time needed to Algorithm 2 is $O(t(B) \cdot n^2)$. $\quad \square$

**Corollary 5** *If $G \in Gen(*, P_3, C_3, C_5)$ then $s(G)$ can be computed in linear time.*

**Proof** By Theorem 6, the components of $\mathcal{D}(G)$ are cliques, stars and $C_5$. For the clique and star components of $\mathcal{D}(G)$, $\zeta(u, v) = \langle 0, 0 \rangle$ for each stretch-pair $u, v$ in a component, obviously computable in constant time. For the $C_5$ components $\zeta(u, v)$ is equal to $\langle 0, 0 \rangle$ or to $\langle 1, 0 \rangle$, but being the vertices at most five, $O(t(B))$ is constant, too. Then the values $a$ and $e$ are computable in constant time, whereas $b, c, f$ are computable in linear time in the size of the number of children of $B$. The value $d$ can be also computed in linear time in the size of the number of children. In fact if the component is a $C_5$ component the number of children is at most five and then the total time is constant. If $B$ is a star or a clique, since $\zeta_B(m_i, m_j)$ is always $\langle 0, 0 \rangle$ then the maximum value is given by the sum of the two maximum values for $second(Z(T, B_i))$ and $second(Z(T, B_j))$, and this can be computed in linear time in the number of children of $B$. Then, since the total number of children is bounded by $O(n)$ and $\mathcal{D}(G)$ can be computed in linear time, then the overall time for Algorithm 2 when $G \in Gen(*, P_3, C_3, C_5)$ is linear in the size of $G$. $\quad \square$

It is not difficult to modify Algorithms 1 and 2 to return also a pair of vertices $u, v$ such that $s(G) = s_G(u, v)$ and to compute the lengths of the longest and shortest paths connecting them, given by $\eta^{-1}(first(Z(T(G), B)))$.

## OTHER COMBINATORIAL PROBLEMS

By using results in *Rao (2008b)*, in this section we show that several basic combinatorial problems can be efficiently solved in the graph class $Gen(*; P3, C3, C5)$. We first provide the definitions of such problems and then we give the results.

Given any graph $G = (V, E)$ and a weight function $w : V \rightarrow \mathbb{N}$, the pair $(G, w)$ is called *weighted graph*. For any subset $V' \subseteq V$, $w(V') = \sum_{v \in V'} w(v)$. Unlike a set, a *multiset* $\langle v_1, v_2, \ldots, v_k \rangle$ allows for multiple instances for each of its elements.

- INDEPENDENT NUMBER: an *independent set* of $G = (V, E)$ is a subset $S \subseteq V$ of vertices of $G$ no two of which are adjacent (i.e., $(u, v) \notin E$ for every $u, v \in V,$). The *weighted independent*

*number* of a weighted graph $(G, w)$ is the maximum weight of an independent set of $G$. It is usually denoted by $\alpha_w(G)$.

- CLIQUE NUMBER: The *weighted clique number* $\omega_w(G)$ of a weighted graph $(G, w)$ corresponds to the maximum weight of a clique of $G$.

- CHROMATIC NUMBER: A proper coloring of a graph $G = (V, E)$ is an assignment of colors to the element of $V$ so that no two adjacent vertices have the same color. The *chromatic number* of $G$ is the minimum number of colors required in a proper coloring; it is denoted $\chi(G)$. It can be observed that if $G$ admits a proper $k$-coloring, then $V$ can be partitioned into $k$ stable sets (i.e., color classes) $V_1, V_2, \ldots, V_k$. Given a weighted graph $(G, w)$, a $k$-coloring of $G$, and the corresponding color classes $V_1, V_2, \ldots, V_k$, we define $w(V_i) = \max\{w(v) | v \in V_i\}$ to be the weight of a color class (or simply the weight of a color). The *weighted coloring* problem is to find a proper $k$-coloring of vertices of $G$ so as to minimize the sum of the weights of the color class, i.e., find a coloring that minimizes $\sum_{i=1}^{k} w(V_i)$. This value is called the *weighted chromatic number* of $(G, w)$.

- DOMINATION NUMBER and its variants: For a graph $G = (V, E)$, a subset $S \subseteq V$ is said to be a *dominating set* of $G$ if $N_G[S] = V$. A dominating set of smallest size is called a *minimum dominating set* and its size is known as the *domination number* $\gamma(G)$. The concept of domination simply extends to weighted graphs, and the *weighted domination number*, denoted by $\gamma_w(G)$, corresponds to the minimum weight of a dominating set. In the literature there exists some variants of the concepts of dominating set:

  - A set $S \subseteq V$ is a *connected dominating set* if $S$ is a dominating set and $G[S]$ is connected; the *weighted connected domination number* is denoted by $\gamma_{cw}(G)$ and corresponds to the minimum weight of a connected dominating set of $G$.
  - A set $S \subseteq V$ is a *dominating clique* if $S$ is a dominating set and a clique; the *weighted dominating clique number* is denoted by $\gamma_{clw}(G)$ and corresponds to the minimum weight of a dominating clique of $G$.
  - A set $S \subseteq V$ is a *independent dominating set* if $S$ is both a dominating set and an independent set; the *weighted independent domination number* is denoted by $\gamma_{iw}(G)$ and corresponds to the minimum weight of an independent dominating set of $G$.
  - A set $S \subseteq V$ is a *total dominating set* if $S$ is a dominating set and $G[S]$ has no isolated vertex; the *weighted total dominating number* is denoted by $\gamma_{tw}(G)$ and corresponds to the minimum weight of a total dominating set of $G$.

- CLIQUE-WIDTH: Let $k$ be a positive integer. A $k$-graph is a graph whose vertices are labeled by integers from $\{1, 2, \ldots, k\}$. We consider an arbitrary graph as a $k$-graph with all vertices labeled by 1. We call the $k$-graph consisting of exactly one vertex $v$ (say, labeled by $i \in \{1, 2, \ldots, k\}$) an initial $k$-graph and denote it by $i(v)$. The clique-width $cwd(G)$ of a graph $G$ is the smallest integer $k$ such that $G$ can be constructed from initial $k$-graphs by means of repeated application of the following three operations:
  1. Disjoint union (denoted by $\oplus$);
  2. Relabeling: changing all labels $i$ to $j$ (denoted by $\rho_{i \to j}$);
  3. Edge insertion: connecting all vertices labeled by $i$ with all vertices labeled by $j$, $i \neq j$ (denoted by $\eta_{i,j}$ or $\eta_{j,i}$) - already existing edges are not doubled.

A construction of a $k$-graph using the above operations can be represented by an algebraic term composed of $\oplus$, $\rho_{i \to j}$, and $\eta_{i,j}$, $(i,j \in \{1,2,\ldots,k\}$, and $i \neq j)$. Such a term is called a *cwd-expression* defining $G$. A $k$-expression is a cwd-expression in which at most $k$ different labels occur. Thus, the clique-width of a graph $G$ is the smallest integer $k$ such that $G$ can be defined by a $k$-expression (see *Courcelle, Engelfriet & Rozenberg, 1993* for a more formal definition).

- GRAPH ISOMORPHISM: two graphs $G_1 = (V_1, E_1)$ and $G_2 = (V_2, E_2)$ are isomorphic if there exists an "edge-preserving bijection" from $G_1$ to $G_2$, that is a function $f : V_1 \to V_2$ such that $(u,v) \in E_1$ if and only if $(f(u), f(v)) \in E_2$.

Till now we have used the notion of split decomposition according to the "uniqueness" Theorem 2. Conversely, the following result given by M. Rao in (*Rao, 2008b*) uses the "minimal split decomposition" version, in which each component is splitted till to reach its minimal size. So, let $\mathscr{G}_k$, for $k \geq 3$, be the class of graphs for which every prime graph in a minimal split decomposition has at most $k$ vertices.

**Theorem 10** (*Rao, 2008b*) *For any fixed* $k \geq 3$:

- *there is an* $O(n)$ *algorithm to compute the weighted stability number, the weighted cliques number, the domination number and its variants respectively, and a* $O(n^3)$ *algorithm to compute the chromatic number of graph in the class* $\mathscr{G}_k$, *if a minimal split decomposition tree is given with the graph;*
- *the clique-width of graphs in the class* $\mathscr{G}_k$ *is bounded by* $2k+1$.

Accordingly, the following results concerning the new class $Gen(*; P_3, C_3, C_5)$ can be easily derived:

**Corollary 6** *For any fixed* $k \geq 3$:

- *there is a* $O(n)$ *algorithm to compute the weighted stability, the weighted cliques number, the domination number and its variants respectively, and a* $O(n^3)$ *algorithm to compute the chromatic number of graph in the class* $Gen(*; P3, C3, C5)$, *if a minimal split decomposition tree is given with the graph;*
- *the clique-width of graphs in the class* $Gen(*; P3, C3, C5)$ *is bounded by 11.*

The next theorem still use the the "uniqueness" version of split decomposition. Let $\mathcal{C}$ the class of graphs for which the isomorphism problem can be solved in time $O(t)$.

**Theorem 11** *If* $G_1$ *and* $G_2$ *are graphs such that the components of both* $\mathcal{D}(G_1)$ *and* $\mathcal{D}(G_2)$ *are in* $\mathcal{C}$, *then the isomorphism between* $G_1$ *and* $G_2$ *can be tested in time* $O(\max\{t, n+m\})$.

**Proof** By Theorem 2, any graph $G$ can be transformed into a unique decomposition tree $T$. By using the algorithm in *Dahlhaus (2000)*, such a tree can be computed in linear time. Hence, testing the isomorphism between $G_1$ and $G_2$ corresponds to testing the isomorphism between the decomposition trees $T_1$ and $T_2$. It is a well known result that the isomorphism problem for trees can be solved in linear time (*Aho, Hopcroft & Ullman, 1974*), and it is easy

to see that such a basic result can be extended to decomposition trees once the isomorphism between vertices of $T_1$ and $T_2$ (i.e., the isomorphism between components of $\mathcal{D}(G_1)$ and $\mathcal{D}(G_2)$) can be performed efficiently. Since components of $\mathcal{D}(G_1)$ and $\mathcal{D}(G_2)$ belong to $\mathcal{C}$, then the isomorphism between $G_1$ and $G_2$ can be tested in time $O(\max\{t, n+m\})$. $\quad\square$

**Corollary 7** *In the class* $Gen(*; P3, C3, C5)$, *the isomorphism problem can be solved in linear time.*

**Proof** By Theorem 6, each component in the split decomposition of graphs in $Gen(*; P3, C3, C5)$ is a star, a clique, or a cycle $C_5$. Then, by Theorem 11, the isomorphism problem in $Gen(*; P3, C3, C5)$ can be solved in linear time. $\quad\square$

## CONCLUSION

In this work we introduced the graph class denoted as $Gen(*; P_3, C_3, C_5)$ and containing all graphs that can be generated by means of split composition using path $P_3$, cycle $C_3$, and any cycle $C_5$ as components. This new graph class extends the well known class of distance-hereditary graphs, which corresponds to $Gen(*; P_3, C_3)$.

For this new class we provided efficient algorithms for several basic combinatorial problems: recognition, stretch number, stability number, clique number, domination number, chromatic number, and graph isomorphism. We also proved that graphs in the new class have bounded clique-width. All these results have been obtained by exploiting in some way the generative definition of each graph belonging to the class according to the classical decomposition approach: the problem at hand is first solved in each component and then the solutions obtained in each component are composed/manipulated to get the solution for the whole graph.

A first possible extension of this work could be to investigate in the class $Gen(*; P_3, C_3, C_5)$ other combinatorial problems that have been solved in the class of distance-hereditary graphs.

A more interesting direction could be that of extending to the whole class sDH(2) the approach used here for solving problems in in the class $Gen(*; P_3, C_3, C_5)$. In particular, we have shown that $Gen(*; P_3, C_3, C_5)$ is a subclass of sDH(2), i.e., each graph in $Gen(*; P_3, C_3, C_5)$ has stretch number less than two. Unfortunately, $Gen(*; P_3, C_3, C_5)$ is a proper subclass of sDH(2), in particular we observed that there are prime graphs in sDH(2) of arbitrary size. This implies that the class sDH(2) cannot be characterized by using the split decomposition, as instead holds for both $Gen(*; P_3, C_3)$ and $Gen(*; P_3, C_3, C_5)$. Since we have shown that the subgraphs forming the "core" of any graph $Gen(*; P_3, C_3, C_5)$ is fully split-decomposable (cf. results in 'Structural Properties About Graphs in sDH(2)'), it would be interesting (*i*) to study how such "core" graphs are joined each other in graphs belonging to sDH(2), and (*ii*) to study whether there exists a sort of "extended split-decomposition" that can be used to characterize graphs in sDH(2). Concerning the latter, observe Fig. 10A: there, a split is missing and hence the graph is prime. If we added two arcs among the four vertices forming the square, then the graph would have the requested split. This suggest to allow an extended split to exists also when some of

the requested edges are missing and, at the same time, this lack of edges creates cycles of bounded size (e.g., cycle no larger than $C_4$). Of course, the existence of a such desired generalized split-decomposition would imply the study of an efficient algorithm for its computation.

## ACKNOWLEDGEMENTS

A preliminary extended abstract appeared in the Proceedings of the 8th Annual Conference on Theory and Applications of Models of Computation (TAMC), 2011 (*Cicerone, 2011b*).

### Funding
The authors received no funding for this work.

### Competing Interests
The authors declare there are no competing interests.

### Author Contributions
- Serafino Cicerone and Gabriele Di Stefano conceived and designed the experiments, performed the experiments, analyzed the data, performed the computation work, prepared figures and/or tables, authored or reviewed drafts of the paper, and approved the final draft.

### Data Availability
The algorithms 1 and 2 are available in the Supplementary File.

### Supplemental Information
Supplemental information for this article can be found online at http://dx.doi.org/10.7717/peerj-cs.627#supplemental-information.

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
