# Peer review of "Getting new algorithmic results by extending distance-hereditary graphs via split composition"

_PeerJ Computer Science, doi:10.7717/peerj-cs.627_

## Round 0.1 · original submission · Major Revisions

Please pay special attention to the comments of Reviewers #2 concerning the contribution of the current work compared to the TAMC conference paper.

Reviewer 1 ·

Basic reporting

The paper is well written and well presented.

Experimental design

The paper fit in the scope of this journal.

Validity of the findings

The results are interesting and as far as I can tell seem valid to me.

Additional comments

This paper deals with a generalization of distance hereditary graphs (DHG for short) and
some structural properties.

A graph is distance hereditary if the distance between any two pair of vertices
is the same in the graph and all of its induced subgraph (until deconnection).

Along the years several characterizations of this class were obtains. Two of them
are particularly relevant in the context of this paper, the first one is the existence
of construction scheme. A graph is DHG if it can be obtained from a single vertex
by adding successively, either a pendant vertex, or true or false twin. The other one,
is the fact that DHG are completely decomposable with respect to split decomposition.

In this article, the authors generalizes the class of DHG by considering this two characterizations.
Namely, they put in use the split Composition operation. DHG, can be obtained by using split
composition, using as key elements P_3 and C_3. In their new generalization they add in the
list of key element the cycle on five vertices: C_5. By doing so, it allows to generalizes the
class.

Then they prove a series of structural results of the newly introduced class. They first shot that
the class is an hereditary class. Then they prove that this class also generalize the class of DHG
with respect to the metric aspect; namely they prove that the shortest distance between any two pair of vertices
in the induced subgraphs is at most twice the shortest distance in the original graphs.

Then they prove how to compute the stretch number for graphs in this class. To do so they exploit the structure
of the split decomposition of the graphs.

Finally they obtain, using a framework developped by M. Rao. The framework allows to solve in polynomial
time a list of optimization problems, by considering the split decomposition. In this
class, since the prime nodes are of constant size. The framework can be successfully applied.
The also prove that the class considered is of bounded cliquewidth.

The paper contains interesting results and concepts.

·

Basic reporting

This paper is an extended version of the following TAMC conference paper published in 2011 by one of the authors:

Cicerone, Serafino. "Using split composition to extend distance-hereditary graphs in a generative way." International Conference on Theory and Applications of Models of Computation. Springer, Berlin, Heidelberg, 2011.

Although it is relatively common for computer scientists to publish full papers that extend previously published conference papers, the extended versions are expected to have enough extra material to warrant an additional publication. Also, they are usually published within 1-2 years after the conference.

The introductory part of the paper under consideration leaves an impression that its main contribution is the detailed proofs of the statements from the 2011 paper. Indeed, lines 89-96 list the exact same results as the 2011 paper. The paper claims to introduce a new class of graphs, which certainly is not the case as the class was introduced (in the exact same way) in the 2011 paper.

My main concern though is the motivation. Why is the class Gen(*; P3, C3, C5) worth introducing and studying? The only sentence that addresses this issue is in line 82: "This consideration leads to redefining the main objective". I don't see, however, how the (somewhat natural) hierarchy DH(k) can be redefined using a single new class.

This concern of mine seems to be shared by the community -- the original 2011 paper has not received any citations (according to Google Scholar) in almost ten years since it was published.

Experimental design

This paper does not contain any experiments.

Validity of the findings

I have not checked all proofs very carefully (due to my concerns raised above) but those that I did look correct.

·

Basic reporting

The paper seems very well written. I could not find any typo's or language mistakes. The presentation is clear and the references are stated properly. The figures are well done and the structure of the article is professional.

Experimental design

The problem studied by the authors is definitely interesting for the graph theory community. The authors define properly the problem and all the preliminary notions. Also, present a nice survey of related results.

Validity of the findings

The proofs presented in the paper seem correct to the best of my knowledge. However, due to limited review time, my review has low confidente. For such a long theoretical paper a more thorough review is needed.

Additional comments

In the paper "Getting new algorithmic results by extending distance-hereditary graphs via split composition" the authors introduce a new distance hereditary graph class Gen(*;P3;C3;C5) that is obtained by applying the split operation recursively starting from P3, C3 and C5. The authors show that the recognition problem for this new class can be solved in linear time. The recognition problem is: given a graph G, decide whether G belongs to Gen(*;P3;C3;C5). Then, they show some properties about graphs in sDH(2) (the graphs with stretch number less than 2) that are used to provide a linear time algorithm for computing the stretch number in any graph belonging to the class Gen(*;P3;C3;C5). Finally they show how known results can be exploited to solve some other combinatorial problems in the new class.


The paper is an extended version of a paper published in TAMC 2011. I found the paper interesting and I believe that is should be accepted for publication. The results are clearly of interest of the graph theory community. The paper seems well written and the authors put a lot of effort to properly define all the notions used. Moreover, I appreciate that the authors provide some intuition and present the high level approach before digging into details. However, due to the short review time, I was not able to check the proofs in detail.

·

Basic reporting

The authors address possible connections between two generalizations of distance-hereditary graphs. One such generalization, rather natural, is the increasing hierarchy of classes DH(k), that contains all graphs whose all induced subgraphs are k-spanners of their vertex-set (w.r.t. the distances in the original graph). The other, more algorithmic, is based on split decomposition.

For that, a superclass of distance-hereditary graphs is defined and studied, based on split decomposition. The full link between the new class and distance-hereditary graphs is partly based on some properties of the latter, which the authors claim to be new.

The authors then prove that every graph in the new class (hereafter called C) is in DH(k) for some k < 2. Let us call sDH(2) the union of all the DH(k) for k < 2. Unfortunately, the authors observe that their approach with split decomposition cannot lead to a nice generation scheme for sDH(2) (in the sense that they are such graphs that are prime for split decomposition and of arbitrary size). However, they did prove that every graph of sDH(2) has some induced subgraph which belongs to the newly defined class C. Furthermore, this induced subgraph, called a core in the paper, is some sort of minimal certificate for the stretch of the graph (the stretch is the least k such that the graph is in DH(k) ).

The main algorithmic result of the paper is a linear-time algorithm for computing the stretch of any graph in C. Finally, the authors observe that due to a very general framework of Rao, several classic combinatorial (NP-hard) problems can be solved in polynomial time on their class C. In particular, all graphs in C have bounded clique-width.

Experimental design

Does not apply.

Validity of the findings

On the negative side, it should be noted that the new results on distance-hereditary graphs that are presented at the beginning of the paper stay very close to previous studies.

The result that every graph in sDH(2) has a ``core'' induced subgraph in the newly defined class C is somehow surprising. The proof is rather tedious, with a lot of intermediate calculations, but I found it very insightful about the internal structure of the graphs in sDH(2).

Regarding the main algorithmic result (computation of the stretch number within C). This result shows the tractability of the stretch number beyond the trivial case of distance-hereditary graphs (for which it equals one). Recall that on general graphs, this problem is NP-hard.The algorithm in itself is simple (dynamic programming on a split decomposition tree), but it requires a very ingenious way to store the output of subproblems. Such techniques might find applications elsewhere, for other resolution techniques based on split decomposition.

Finally, I want to mention that the authors should revise their abstract (and possibly some other parts of the paper) because they did NOT provide an algorithm for computing the clique-width of the graphs in C. They only proved it was bounded.

Additional comments

Overall, I enjoyed the paper and I recommand it for acceptance.

MINOR COMMENTS:

*p18 (14/26), l408, case 2: u1 is repeated twice in the definition of the path

*p22 (18/26), l468: typo: (y,a) instead of (y,b)

*In the legend of Figure 14: "whereas (m2,v′) it is not." => remove ``it''

*p24 (20/26), l502: typo: the the sequences

*The proof of Theorem 9 invokes the result of Theorem 6. However, the latter is only for Gen(*,P3,C3,C5), which is a more restricted case than the one discussed in the statement of Theorem 9.
This is probably just a bad copy/paste: this property is never used in the proof.

*Corollary 5 could be simplified by using (as Rao and others did) a split decomposition where all stars and cliques are further decomposed in C3 and P3.

---

## Round 0.2 · accepted · Accept

As far as I can see, the authors have adequately responded to the comments raised by the reviewers.

·

Basic reporting

no comment

Experimental design

no comment

Validity of the findings

no comment

Additional comments

no comment